

# Minimizing aerosol effects on the OMI tropospheric NO₂ retrieval – An improved use of the 477 nm O₂-O₂ band and an estimation of the aerosol correction uncertainty

Julien Chimot[1,a], J. Pepijn Veefkind[1,2], Johan F. de Haan[2], Piet Stammes[2], and Pieternel F. Levelt[1,2]

[1]Department of Geoscience and Remote Sensing (GRS), Civil Engineering and Geosciences, Delft University of Technology (TU Delft), the Netherlands
[2]Royal Netherlands Meteorological Institute, De Bilt, the Netherlands
[a]now at European Organisation for the Exploitation of Meteorological Satellites (EUMETSAT), Darmstadt, Germany

*Correspondence to:* Julien Chimot (Julien.Chimot@eumetsat.int)

**Abstract.** Global mapping of satellite tropospheric NO₂ vertical column density (VCD), a key gas in air quality monitoring, requires accurate retrievals over complex urban and industrialized areas. The high abundance of aerosol particles in regions dominated by anthropogenic fossil fuel combustion, mega-cities and biomass burning affects the space-borne spectral measurement. Minimizing the tropospheric NO₂ VCD biases under such conditions are one of the main challenges for the retrieval

from air quality satellite instruments. In this study, reference Ozone Monitoring Instrument (OMI) DOMINO-v2 product was reprocessed over cloud-free scenes, by applying new aerosol correction parameters retrieved from the 477 nm O₂-O₂ band, over east China and South America for 2 years (2006-2007). These new parameters are based on two different and separate algorithms developed during the last two years in view of an improved use of the 477 nm O₂-O₂ band: 1) the updated OM-CLDO2 algorithm which derives improved effective cloud parameters, 2) the aerosol neural network (NN) giving explicit

aerosol parameters by assuming a more physical aerosol model. The OMI aerosol NN is a step ahead to OMCLDO2 by retriev-ing primarily an explicit aerosol layer height (ALH), and secondly an aerosol optical thickness $\tau$ for cloud-free observations. Overall, it was found that all the considered aerosol correction parameters reduce the biases identified in DOMINO-v2 over scenes in China with high aerosol abundance and scattering particles: e.g. from $[-20{:}-40]\%$ to $[0{:}20]\%$ in summertime. The use of the retrieved OMI aerosol parameters leads in general to a more explicit aerosol correction and higher tropospheric NO₂

VCD values, in the range of $[0{:}40]\%$, than from the implicit correction with the updated OMCLDO2. This number overall represents an estimation of the aerosol correction strategy uncertainty nowadays for tropospheric NO₂ VCD retrieval from space-borne visible measurements. The explicit aerosol correction theoretically includes more realistic aerosol multiple scat-tering and absorption effects, especially over scenes dominated by strongly absorbing particles, where the correction based on OMCLDO2 seems to remain insufficient. However, the use of ALH and $\tau$ from the OMI NN aerosol algorithm is not a straight-

forward operation and future studies are required to identify the optimal methodology. Several elements to be considered are recommended in this paper. Overall, we demonstrate the possibility to apply a more explicit aerosol correction by considering aerosol parameters directly derived from the 477 nm O₂-O₂ spectral band, measured by the same satellite instrument. Such an approach can, in theory, easily be transposed to the new-generation of space-borne instruments (e.g. TROPOMI on-board



Sentinel-5 Precursor), enabling a fast reprocessing of tropospheric $NO_2$ data over cloud-free scenes (cloudy pixels need to be filtered out), as well as for other trace gas retrievals (e.g. $SO_2$, HCHO).

## 1   Introduction

Long-time series of UV-visible (UV-vis) satellite measurements are a great asset for monitoring the distribution and evolution of pollutants such as $NO_2$, HCHO, or $SO_2$ and aerosol particles in the troposphere. With the forthcoming new generation of sensors like TROpospheric Ozone Monitoring Instrument (TROPOMI) on-board Sentinel-5-Precursor (Veefkind et al., 2012), Sentinel-4-UVN, and Sentinel-5-UVNS within the Copernicus programe (Ingmann et al., 2012), they will become an important tool for verifying the effectiveness of implemented technology to protect environment and health population (Duncan et al., 2016). While the last generation of space instruments have had a pixel size between 13 x 24 $km^2$ for the Ozone Monitoring Instrument (OMI) or 80 x 40 $km^2$ for the Global ozone Monitoring Experiment (GOME-2), the new generation have smaller pixel sizes (about 7 x 3.5 $km^2$ for TROPOMI), allowing air quality mapping of complex urban and city areas. This is also expected to reduce the probability of cloud contamination. However, the significant probability of aerosol contamination in areas such as India, China or regions dominated by biomass burning episodes will likely remain, or may even increase.

OMI is the Dutch Finnish push-broom spectrometer flying on the National Aeronautics and Space Administration (NASA)'s Earth Observation Satellite (EOS)-Aura satellite since July, 15th 2004. Its Sun-synchroneous orbit has a local equator crossing-time of approximately 13:40h. Operational tropospheric $NO_2$ products derived from the visible backscattered spectral light (405-465 nm) such as the OMI DOMINO-v2 (Boersma et al., 2011) or the very recent Quality Assurance for Essential Climate Variables (QA4ECV) are nowadays a reference. Their related global mapping of tropospheric $NO_2$ concentrations have been used by many air quality research studies focusing on $NO_x$ emissions, secondary pollutant formation, as well as tropospheric $NO_x$ chemistry and transport: e.g. (Curier et al., 2014; Reuter et al., 2014; Ding et al., 2015).

A critical element for an accurate tropospheric $NO_2$ vertical column density (VCD) retrieval is our capability to reproduce the average light path along which the photons travelled before being detected by the satellite sensor in the visible spectral window. In particular, scattering induced by atmospheric aerosol particles over cloud-free scenes are known to lead to very complex light path. Because they are emitted by the same sources, high $NO_2$ and aerosol concentrations are often spatially correlated (Veefkind et al., 2011). Therefore, the presence of aerosol needs to be properly addressed in the retrieval algorithms. In the frame of tropospheric $NO_2$ retrievals from visible spectral measurements which use the differential optical absorption spectroscopy (DOAS) approach, the aerosol correction has to be done on the air mass factor (AMF): a unitless number representative of the length of the average light path.

Over cloud-free scenes, a full explicit aerosol correction ideally requires a comprehensive set of parameters describing aerosols: i.e. optical properties such as the single scattering albedo $\omega_0$, scattering phase function, load, microphysical parameter such as the size, and vertical distribution (Martin et al., 2003; Leitão et al., 2010; Bousserez, 2014). Among all these variables, many studies emphasized the importance of the aerosol layer height (ALH) knowledge (Leitão et al., 2010; Castellanos et al.,





2015; Chimot et al., 2016). Assuming no aerosol correction, i.e. aerosol-free scene (Richter and Burrows, 2002), would clearly create large biases on the OMI tropospheric $NO_2$ retrievals (Chimot et al., 2016).

There are basically two strategies for applying the aerosol correction in the AMF: 1) either by considering external data, or 2) by using the available particle parameters that can be simultaneously derived from the same UV-visible spectral space-
borne measurement. Studies that reprocessed DOMINO-v2 dataset using external data usually combined atmospheric transport model outputs such as GEOS-Chem in the Peking University OMI $NO_2$ (POMINO) (Lin et al., 2014, 2015), or observations issued from different satellite platforms such as the Cloud-Aerosol Lidar with orthogonal Polarization (CALIOP) (Castellanos et al., 2015), or even both combined together (Liu et al., 2018). Resulting changes mostly occurred in case of high aerosol pollution ($\tau(550\,nm) > 0.8$) with increased or decreased tropospheric $NO_2$ VCDs depending on the geophysical conditions and
aerosol properties and distributions. However, the resulting AMF computation becomes dependent on these data sources, their quality and the possibility (or not) to combine them altogether. In general, spatial and temporal co-registration between the different instruments or due to different resolution between the observation pixel and the model grid cell may become an issue. In the frame of an operational processing, it is generally preferred to maximize the exploitation of the spectral measurement acquired by a same instrument representative of the considered observation pixel. One of the main reasons is the need to have
an indication on particle height representative of the average light path associated with every single OMI field of view (FOV). Such an information is generally not easily and directly available from an external source. Exploitation of the 477 nm $O_2$-$O_2$ absorption for aerosol retrieval is very promising. It is not only measured by OMI, but also by GOME-2, TROPOMI, Sentinel-4-UVN and Sentinel-5-UVNS. Several studies based on ground-based and satellite instrument have demonstrated its relative high sensitivity to aerosols, in particular to ALH (Wagner et al., 2004; Castellanos et al., 2015; Park et al., 2016; Chimot et al.,
20  2016, 2017).

Because of the difficulty to easily distinguish clouds from aerosols, and to identify the right aerosol model to use, it has always been preferred to retrieve effective clouds assuming a Lambertian and opaque reflector model (Acarreta et al., 2004; Stammes et al., 2008) and consequently to compute the resulting troposphere $NO_2$ AMF for all the OMI scenes, regardless of the presence of clouds, aerosols or both together. Such correction is historically named an "implicit aerosol correction"
(Boersma et al., 2004, 2011). Chimot et al. (2016) clearly demonstrated that, in spite of its implicit nature, such a correction allows to mitigate biases on the OMI DOMINO $NO_2$ product over cloud-free scenes compared to an aerosol-free pixel assumption. However, limitations were identified: 1) a numerical artifact is present due to a too coarse sampling employed in the OMI cloud look-up-table (LUT) leading then to a strong underestimation on the OMI tropospheric $NO_2$ VCD over scenes with strong aerosol load ($\tau(550\,nm) \geq 0.6$) and particles located at high altitude, 2) the Lambertian cloud model, in spite of
its benefits, remains somehow too simple and likely does not fully reproduce all the multiple scattering effects inherent to aerosol particles. The OMI effective cloud algorithm was then updated in order to remove these numerical artifacts (Veefkind et al., 2016). It also includes many additional relevant improvements of the OMI 477 nm $O_2$-$O_2$ band. But its impact on the correction for aerosols has not yet been evaluated.

To move one step further, Chimot et al. (2017) developed a novel machine learning algorithm, based on the neural network
(NN) technique, that allows to retrieve ALH together with aerosol optical thickness (AOT) $\tau$ from the same OMI 477 nm $O_2$-$O_2$





spectral band over cloud-free scenes. These retrievals were performed on various cases both over land and sea and compared with reference CALIOP observations and related climatology (Chimot et al., 2017, 2018). They benefit from a strong synergy with MODIS on-board the NASA Aqua platform flying together with Aura in the same NASA A-Train constellation, in order to identify the cloud-free scenes and to better constrain the ALH retrieval quality. For such a purpose, the 477 nm $O_2$-$O_2$

band represents some advantages compared to the more traditional $O_2$-A band at 760 nm which is not measured by OMI: 1) it is spectrally closer to the $NO_2$, HCHO, or $SO_2$ absorption features, 2) it has a wider spectral range although weaker signal leading to high sensitivities in case of high aerosol loading, and 3) it has less radiative transfer challenges arising from strong absorption lines. Moreover, the NN technique development allows a very fast OMI data processing which is an important requirement within an operational environment. The aerosol retrievals performed with this algorithm are expected to lead to

an explicit aerosol correction over cloud-free scenes by using the OMI 477 nm $O_2$-$O_2$ measurement simultaneously acquired with the 405-465 nm $NO_2$ band.

    This paper aims to evaluate the benefits of our improved use of the OMI 477 nm $O_2$-$O_2$ band for correcting aerosol effects in tropospheric $NO_2$ VCD retrieval from the same visible observation. We evaluate the potential of directly using the OMI NN aerosol ALH (and $\tau$) in view of an explicit correction. We also assess the expected changes in the implicit aerosol correction

based on the improved OMI effective cloud algorithm. To compare the different aerosol correction strategies, we reprocessed 2 years (2006-2007) of the DOMINO-v2 data over different areas and seasons, dominated by different types of pollution episodes and thus, $NO_2$ and aerosol sources: 1) the large urban and industrialized east China region dominated by a mix of continental polluted particles and dust particles in summer time (June-July-August), 2) the same area essentially dominated by continental polluted particles in winter time (December-January-February) times, and 3) South America during the biomass burning season

associated with heavy absorbing aerosol emission (August-September). Given all the aerosol corrections available from on our improved use of the OMI 77 nm $O_2$-$O_2$ band, their comparison in this paper gives an estimation of the aerosol correction uncertainty in the OMI tropospheric $NO_2$ VCD retrieval. Sections 2 and 3 describe the algorithms and the reprocessing methodology. Section 4 evaluates the results of the applied aerosol corrections in the reprocessed tropospheric $NO_2$ retrievals. To complete the analyses, Sect. 5 includes specific discussions based on reference simulations to better understand the behaviour of the

new OMI tropospheric $NO_2$ VCD. Similarly to Chimot et al. (2016), the advantage of such simulations is to determine, on well controlled cases, the expected new biases on the OMI tropospheric $NO_2$ VCD and to identify the key geophysical factors driving them. At the end, in Sect. 6, we discuss about the benefits and challenges of each aerosol correction.

## 2   The OMI $O_2 - O_2$ algorithms

### 2.1   $O_2 - O_2$ DOAS spectral fit

In this paper, both effective cloud and aerosol algorithms are based on the same OMI 477 nm $O_2 - O_2$ spectral band. More specifically, they use the continuum reflectance $R_c(475$ nm$)$ and the $O_2$-$O_2$ slant column density (SCD) $N^s_{O_2-O_2}$. These variables are derived from the DOAS spectral fit approach, which is then a prerequisite before applying either the OMI cloud look-up-table (LUT) (see Sect. 2.2) or the aerosol neural networks (see Sect. 2.3).





The DOAS method is a specific spectral fit approach following basic principle of absorption spectroscopy employed for UV and visible absorbing trace gases. The various DOAS techniques rely on the same key concept: a simultaneous fit of several trace gas slant column densities from the fine spectral features due to their absorption (i.e. the high frequency part) present in passive UV–visible spectral measurements of atmospheric radiation (Platt and Stutz, 2008). The assumed Beer-Lambert (or Bouguer-Lambert) law describes the light attenuation as a function of the travelled distance in the atmosphere, gas concentration and its spectral absorption intensity. It is commonly employed for absorption spectroscopy analyses of $NO_2$, $SO_2$, HCHO and $O_3$ from the OMI, TROPOMI, GOME, GOME-2 and SCIAMACHY sensors: e.g. (Boersma et al., 2011; De Smedt et al., 2017). The spectral fit is achieved within a predefined spectral window and the slant column density $N_x^s$ is defined as the column density of a trace gas absorber $x$ along the average light path travelled by the detected photons from the Sun through the atmosphere, surface and back to the satellite sensor.

Here, the OMI 477 nm $O_2$-$O_2$ DOAS fits together the absorption cross-section spectrum of $O_2$-$O_2$ with a first-order polynomial over the (460–490 nm) spectral band (Acarreta et al., 2004; Veefkind et al., 2016). The continuum reflectance $R_c$ at the reference wavelength $\lambda_0 = 475$ nm is the reflectance which would be measured in the absence of $O_2$-$O_2$ in the atmosphere.

In the absence of clouds, both OMCLDO2 and OMI aerosol algorithms rely on how aerosols affect the length of the average light path along which $O_2$-$O_2$ absorbs. $R_c$(475 nm) is known to represent the enhanced scene brightness due to the additional scattering effects induced by the particles. In particular, $R_c$(475 nm) directly increases with increasing AOT $\tau$. This enhancement depends however on aerosol properties as well as the surface albedo (Boersma et al., 2011; Chimot et al., 2016; Castellanos et al., 2015). $N_{O_2-O_2}^s$ is governed by the overall shielding or enhancement effect of the absorption of the photons by the $O_2$-$O_2$ complex in the visible spectral range along the average light path. A reduction of the length of the average light path, i.e. shielding effect, reduces the absorption by $O_2$-$O_2$. The aerosol layer height is the primary driver (Castellanos et al., 2015; Chimot et al., 2016, 2017). An aerosol layer located at high altitudes causes a large shielding effect on the $O_2$-$O_2$ located in the atmospheric layers below, by reducing the amount of photons coming from the top-of-atmosphere and reaching the lowest part of the atmosphere compared to an aerosol-free scene. As a second order effect, aerosol properties such as load and single scattering albedo, and surface reflectance also contribute to $N_{O_2-O_2}^s$.

## 2.2 OMI cloud algorithm OMCLDO2

The OMI cloud algorithm also named OMCLDO2 (Acarreta et al., 2004) derives the effective cloud fraction $c_f$ and cloud pressure $c_p$ assuming the cloud model as an opaque Lambertian reflector with a constant albedo of 0.8 (Stammes et al., 2008) and the independent pixel approximation (IPA) (Martin et al., 2002; Acarreta et al., 2004). The measured reflectance $R$ is formulated as a linear combination of a clear-sky $R_{Clear}$ and a cloudy reflectance $R_{Cloud}$ (Martin et al., 2002; Acarreta et al., 2004):

$$R(\lambda) = c_f \cdot R_{Cloud} + (1 - c_f) \cdot R_{Clear}. \tag{1}$$

A look-up-table (LUT) enables the conversion of $R_c$(475 nm) and $N_{O_2-O_2}^s$ into $c_f$ and $c_p$. It requires knowledge on the surface reflectance and surface pressure in addition to satellite and sun geometry configurations (Acarreta et al., 2004; Veefkind



et al., 2016). Because of the low impact of small clouds on the $O_2$-$O_2$ band, $c_p$ has large uncertainties in case of low $c_f$ (Acarreta et al., 2004). The term "effective" here means that these cloud parameters do not represent actual clouds, but our best explanation of the measured radiance by combining these variables with the assumed approximate model (Sneep et al., 2008; Stammes et al., 2008). Therefore, the retrieved $c_f$ and $cp$ values of each observed scene match with the measurement

summarized by ($R_c$(475 nm)- $N^s_{O_2-O_2}$), such that the (460–490 nm) radiance budget is comprehensively closed (apart from instrument noise). For example, true optically thin clouds will be retrieved as a an opaque and bright Lambertian reflector covering only a small part of the OMI pixel, mostly because of the large assumed cloud albedo value (Veefkind et al., 2016).

The main motivation of this cloud retrieval scheme has been the correction of cloud effects in trace gas retrievals (Stammes et al., 2008). However, this algorithm is actually applied both to cloudy and cloud-free scenes with aerosols, without any prior

distinction. Many studies demonstrated that OMCLDO2 accounts for a large part of aerosol effects by retrieving $c_f$ and $c_p$ (Boersma et al., 2004, 2011; Castellanos et al., 2015; Chimot et al., 2016; Wang et al., 2015). Under these conditions, the OMI cloud parameters become then "more effective" as they do not represent anymore cloud but aerosol effects on the (460–490 nm) radiance. One could claim that OMCLDO2 becomes then an approximate aerosol model, independent of those considered in Sect. 2.3. Chimot et al. (2016) demonstrated how OMCLDO2 responses to aerosols: 1) $c_f$ is mostly driven by $R_c$(475

nm) and increases with increasing aerosol load, regardless of its altitude. Its magnitude is weighted by aerosol properties and surface conditions. 2) $c_p$ represents beforehand the degree of shielding effect applied by aerosols, which results from a complex combination of ALH as a first order, aerosol load $\tau$ and type, surface properties and geometry angles as a second order. A stronger shielding effect leads to a lower $c_p$. In general, over scenes with high $\tau$ values, $c_p$ correlates well with ALH. Furthermore, regardless of true aerosol layer altitude, absorbing particles lead to a decrease of $c_p$, while the presence of more

scattering particles increases $c_p$ values (Castellanos et al., 2015; Chimot et al., 2016).

Veefkind et al. (2016) released a new version of the OMCLDO2 product. The new algorithm, here named OMCLDO2-New, includes several improvements such as a better consistency of gas absorption cross sections with the OMI $NO_2$ retrieval algorithm, outlier removal from the spectral fitting, etc.. However, in the context of the implicit aerosol correction, the expected highest changes come from the higher number of nodes of the OMI cloud LUT. Indeed, the coarse sampling of the OMI

cloud LUT associated with the OMCLDO2-Old version created a numerical artefact: $c_p$ was increasing with decreasing $c_f$ (or aerosol $\tau$) without any physical explanation (Chimot et al., 2016; Veefkind et al., 2016). This has strong impacts in the OMI tropospheric $NO_2$ product, DOMINO v2, in scenes dominated by aerosols (see Sect. 3.3). Furthermore, a temperature correction is implemented in OMCLDO2-New to take into account the density squared dependence of the $O_2$-$O_2$ absorption. Its impact however depends on the temperature conditions and latitude area (Veefkind et al., 2016).

## 2.3   OMI aerosol neural network

The OMI $O_2$-$O_2$ aerosol algorithm relies on a NN multilayer perceptron approach to retrieve primarily the key parameter ALH over cloud-free scenes, but also aerosol $\tau$(550 nm) as a secondary parameter (Chimot et al., 2017, 2018). Since a fine characterization of aerosol vertical profiles cannot be retrieved from OMI UV-visible measurements, they are assumed as one box layer with a constant pressure thickness (100 hPa). ALH is the mid-altitude of this layer in km over sea level but can also





be expressed in pressure. Here, the strategy differs from the OMI effective clouds of Sect. 2.2. The main motivation is to try to reproduce aerosol scattering effects in the visible spectrum via a more explicit aerosol model than the opaque Lambertian reflector.

The particle properties in this layer are homogeneous. Among all the input parameters, ALH retrieval requires an accurate
$\tau$(550 nm) value as input as ALH and $\tau$(550 nm) simultaneously affect $N^s_{O_2-O_2}$ and need to be separated (Chimot et al., 2017). In theory, this source of information may be diverse (e.g. atmospheric models, prior guess, or observations). In practice, MODIS $\tau$ product has systematically been preferred due to its good spatial and temporal collocation with OMI and its recognized high quality. Retrieved OMI $\tau$ may also be used as they come from a same spectral measurement (same instrument). However due to its higher uncertainty compared to MODIS, its use impacts the quality of OMI ALH (Chimot et al., 2017). For OMI $\tau$ retrieval,
$R_c$(475 nm) is considered instead of $\tau$ as prior input. Note that in the next sections, we define $NN_{MODIS}$ when prior MODIS $\tau$ is considered, $NN_{OMI}$ based on the retrieved OMI $\tau$, and $NN_{True}$ when true $\tau$ value is considered for the synthetic cases (see Sect. 3.1).

The training data set was generated by full-physical spectral simulations, assuming explicitly aerosol particles without clouds, generated by the Determining Instrument Specifications and Analyzing Methods for Atmospheric Retrieval (DISAMAR)
software of KNMI (de Haan, 2011). Aerosol scattering phase function $\Phi(\Theta)$ was simulated by the Henyey–Greenstein (HG) function parameterized by the asymmetry parameter $g$, the average of the cosine of the scattering angle (Hovenier and Hage, 1989). Aerosols were specified as standard fine particles with a unique value of the extinction Ångström exponent $\alpha = 1.5$ and $g = 0.7$. They are assumed to fully cover the OMI pixel. To take into account the inaccuracies of the assumed aerosol single scattering albedo $\omega_0$ properties, two training data sets were generated with a different typical value: one with $\omega_0 = 0.95$ and
one with $\omega_0 = 0.9$ in the visible spectral domain. Therefore, two separate OMI ALH NN algorithms have been created, one for each aerosol $\omega_0$ values.

The HG function is known to be smooth and reproduce the Mie scattering functions reasonably well with $g = 0.7$ for most of aerosol types (Dubovik et al., 2002). A similar approach is considered for the operational ALH retrieval algorithms for Sentinel-4 and Sentinel-5 Precursor (Leitão et al., 2010; Sanders et al., 2015; Colosimo et al., 2016; Nanda et al., 2017), and
when applying various explicit aerosol corrections in the tropospheric $NO_2$ AMF calculation over urban and industrial areas dominated by anthropogenic pollution, for instance in east China (Spada et al., 2006; Wagner et al., 2007; Castellanos et al., 2015; Vlemmix et al., 2010).

Similarly to the high $cp$ inaccuracy in case of low $cf$, high ALH bias is expected below a minimum particle load (i.e. threshold of $\tau$(550 nm) = 0.5). This is directly due to the nature of the $O_2$-$O_2$ spectral band. Below this threshold, low amounts
of aerosols have negligible impacts on t$N^s_{O_2-O_2}$.

Over cloud-free scenes, OMI ALH has shown consistent spatial patterns with CALIOP level 2 (L2) ALH over urban and industrial areas in east China, with an uncertainty in the range of [500:700] m and for collocated MODIS scenes with $\tau$(550 nm) $\geq 0.5$ (Chimot et al., 2018). Additional analyses showed that differences between the LIdar climatology of vertical Aerosol Structure for space-based lidar simulation (LIVAS) and 3-year OMI ALH with MODIS $\tau(550m) \geq 1.0$ were in the range of
[180:800] m (Amiridis et al., 2015; Chimot et al., 2017). Finally, Chimot et al. (2018) showed the potential of OMI visible





measurements to observe the height of thick and absorbing aerosol layers released by widespread fire episodes such as in South America. The aerosol model assumptions, in particular $\omega_0$, are the most critical as they may affect ALH retrieval uncertainty up to a maximum of 660 m. An accuracy of 0.2 is necessary on prior $\tau(550\text{ nm})$ information to limit ALH bias close to zero over scenes with $\tau(550\text{ nm}) \geq 1.0$, and below 500 m for $\tau(550\text{ nm})$ values smaller than 1.0.

A summary of all the OMI NN aerosol algorithms as well as related input and output parameters is given in Tab. 1.

## 3    From aerosol impacts to aerosol correction – Methodology

### 3.1    General methodology

OMI tropospheric $NO_2$ data are taken from the DOMINO-v2 dataset (see Sect. 3.3). They are reprocessed by recomputing the AMF (see Sect. 3.2) using the DISAMAR radiative transfer model over cloud-free scenes contaminated by aerosols. This AMF
replaces then the original AMF of DOMINO and is applied to the available $NO_2$ SCD.

The computation of the tropospheric $NO_2$ AMF follows the formulation detailed in Sect. 3.2 and applies either an implicit or an explicit aerosol correction: the implicit correction considers the effective cloud retrievals obtained from OMCLDO2 (cf. Sect. 2.2); the explicit aerosol correction employs aerosol parameters: either OMI ALH and OMI $\tau$ from the OMI aerosol NN (cf. Sect. 2.3), or OMI ALH and MODIS $\tau$. The complementary aerosol parameters (i.e. $\omega_o$, $g$, $\alpha$) follow those specified in the
associated training dataset.

The surface albedo is based on the OMI Lambertian Equivalent Reflectivity (LER) climatology (Kleipool et al., 2008). In DOMINO-v2, this climatology is based on a 3-year OMI time series measurements. However, it evolved since with an extended 5-year OMI time series (Veefkind et al., 2016). This evolved OMI LER is considered for all the tropospheric $NO_2$ VCD reprocessing performed in this study. All the other geophysical parameters associated with DOMINO-v2, such as the
$NO_2$ vertical profile, remain identical.

To identify cloud-free OMI observation pixels with aerosols, a similar strategy as Chimot et al. (2016, 2017) is considered. The DOMINO-v2 $NO_2$ scenes are collected together with the MODIS-Aqua aerosol $\tau(550\text{ nm})$ from the combined Dark Target (DT) and Deep Blue (DB) product of Collection 6 available at the resolution of 10 km (Levy et al., 2013). They are collocated within a distance of 15 km. The probability of cloud-free OMI scene is a priori ensured by the availability of the MODIS
aerosol product with the highest quality assurance flag. In such a case, MODIS Aqua $\tau$ was then exclusively retrieved when a sufficient high amount of cloud-free sub-pixels was available (i.e. at the MODIS measurement resolution of 1 km) (Levy et al., 2013). However, it is well recognized this may be not completely representative for the atmospheric situation of the OMI pixel. Therefore, we added two thresholds for each collocated OMI-MODIS pixel: the geometric MODIS cloud fraction to be smaller than 0.1, and OMI $c_f$ lower than 0.1. Past experiences showed that OMI $c_f$ values in the range of [0.1:0.2] may still
contain clouds (or both clouds and aerosols) (Boersma et al., 2011; Chimot et al., 2016).

Additional synthetic cases analysed in Sect. 4 and 5 are also based on the DISAMAR model, specified in a similar way as the NN training dataset in Sect. 2.3. Either OMCLDO2 or the OMI NN aerosol algorithms are used to determine the expected tropospheric $NO_2$ VCD biases.





## 3.2 Air mass factor computations

The computation of tropospheric $NO_2$ AMF $A_{NO_2}$ is a key step for converting $NO_2$ SCD $N^s_{NO_2}$ into tropospheric $NO_2$ VCD $N^v_{NO_2}$, which represents the number of $NO_2$ molecules $cm^{-2}$ integrated along the vertical direction from the surface $P_0$ to the tropopause $P_{trop}$ pressure. The application of $A^v_{NO_2}$ is crucial to correct of the average light path variability contained in $N^s_{NO_2}$.

$A_{NO_2}$ computation has generally been recognized as the principal source of errors in $N^v_{NO_2}$ determination in areas with a high level of air pollution (Boersma et al., 2007). This was even more emphasized by Lorente et al. (2017) who discussed how AMF structural uncertainty is driven by assumed prior information, and cloud and aerosol correction strategies: up to 42% over polluted regions, and 31% over unpolluted regions.

   In the context of OMI visible spectral measurements, $A_{NO_2}$ is defined as the ratio of the atmospheric SCD and VCD (Boersma

et al., 2011):

$$A^v_{NO_2}(\Psi, \lambda) = N^s_{NO_2}(\Psi, \lambda)/N^v_{NO_2}, \qquad (2)$$

   with $\Psi$ the list of input parameters prerequisite for the radiative transfer model. Note that before performing this conversion, the stratospheric and tropospheric contributions to $N^s_{NO_2}$ must be separated. Therefore, $A_{NO_2}(\Psi, \lambda)$ is only applied to the tropospheric $NO_2$ SCD. The OMI tropospheric $NO_2$ formulation follows Palmer et al. (2001); Boersma et al. (2004) and the

concept of altitude-resolved AMF $a(z)$ (also named block AMF or BAMF) introduced by Palmer et al. (2001); Eskes and Boersma (2003), and then generalised by Wagner et al. (2007); Rozanov and Rozanov (2010); Richter and Wagner (2011). The ratio of $a$ to the total air mass factor $A_{NO_2}$ (deduced from the $NO_2$ shape profile) gives the vertical averaging kernel AK: i.e. the sensitivity of the satellite measurement to each vertical atmospheric layer (Eskes and Boersma, 2003; Richter and Wagner, 2011). Overall, $A^v_{NO_2}$ can then be seen as a unitless number representative of the length of the average light path followed

by the detected photons in the troposphere. It includes then an indication about the sensitivity to the amount of $NO_2$ in the troposphere, larger values indicating a higher sensitivity assuming no change in vertical $NO_2$ profile. Indeed, in those cases, a change in $A^v_{NO_2}$ is directly associated with a change of $a$ at the atmospheric levels where the trace gas is present. The reference wavelength considered in this paper is 439 nm, following the OMI $NO_2$ product (see Sect. 3.3) (Boersma et al., 2011).

   Aerosols may cause either a shielding or an enhancement effect. A shielding effect occurs when the length of the average

light path is reduced leading then to a decrease of $A^v_{NO_2}$. Reciprocally, an enhancement affect results in an increase of $A^v_{NO_2}$ (Leitão et al., 2010; Chimot et al., 2016). Following Eq. 2, any bias in $A^v_{NO_2}$ calculation leads to a direct bias into $N^v_{NO_2}$: same value but opposite sign.

   Note that in the case of real OMI tropospheric $NO_2$ retrievals, a temperature correction is often applied as the temperature of the assumed $NO_2$ absorption cross section, fixed at 221 K, can differ from the actual temperature when deriving $N^s_{NO_2}$. The

correction term is thus implemented in the computation of $A^v_{NO_2}$ such that it represents the ratio of $N^s_{NO_2}$ derived with a $NO_2$ cross section at the real temperature $T$ to the column derived at 221 K. European Centre for Medium-Range Weather Forecasts (ECMWF) temperature fields are used for this correction (Boersma et al., 2004, 2011).

   Computation of $A^v_{NO_2}$ requires accurate knowledge about all the parameters $\Psi$ affecting the optical properties of the atmosphere and the length of the average light path. For an aerosol and cloud-free scene, $\Psi$ generally includes the satellite and





solar geometries, ground-pressure and the surface reflectance. In the presence of clouds and/or aerosols,adequate parameters describing their properties must be added.

### 3.3 OMI tropospheric NO$_2$ dataset – DOMINO v2

DOMINO v2 (Boersma et al., 2011) is a reference worldwide tropospheric NO$_2$ product derived from the OMI visible mea-
surements and can be downloaded from the Tropospheric Emissions Monitoring Internet Service (TEMIS) website (http: www.temis.nl).

Chimot et al. (2016) demonstrated that the implicit aerosol correction in DOMINO-v2 is better than the clear-sky assumption (Richter and Burrows, 2002)), with remaining biases between -20% and -40% on tropospheric NO$_2$ VCD, especially in presence of absorbing particles and for $\tau$(550 nm) $\geq$ 0.5. One the main identified limitation was the coarse sampling of the
OMI cloud LUT nodes used in OMCLDO2 (see Sect. 2.2). The effect of OMCLDO2-New version on the implicit aerosol correction have not yet been analyzed.

To our knowledge, no reprocessing has yet been done by applying an explicit aerosol correction based on (nearly) explicit aerosol parameters that are retrieved from the OMI 477 nm O$_2$-O$_2$ spectral band. Thus, the use of OMI ALH and $\tau$ parameters from Sect. 2.3 is a first attempt to apply a (nearly) explicit aerosol correction in $A_{NO_2}^v$ computation by using visible spectral
measurements acquired by a same sensor.

DOMINO has recently evolved through the Quality Assurance for Essential Climate Variables (QA4ECV) project (www. qa4ecv.eu) which aims to address reliable and fully tracable quality infomration on some of the "essential climate variables" (ECVs) , such as tropospheric NO$_2$, as defined by the Global Climate observing System (GCOS) (Lorente et al., 2017). This reprocessing contains numerous changes in the complete chain of retrieval, from the calibrated spectrum, spectral fitting with
DOAS, to the AMF computation and all the ancillary dataset. This new generation of product is expected to represent one of the best NO$_2$ dataset. Since the reprocessing products of QA4ECV is still under thorough validation and was not completely available at the time of this paper (and its technical work), and given our specific objective focused on the aerosol scattering and absorption correction by using information from the O$_2$-O$_2$ spectral band, the last version of DOMINO (v2) has been preferred.
The OMI cloud algorithm configuration used at the time of DOMINO-v2, and its comparison with the other algorithms are summarized in Tab. 1.

## 4 Results of reprocessing OMI NO$_2$ and O$_2$ − O$_2$ products

All the OMI tropospheric NO$_2$ reprocessings achieved here are based on the OMI cloud and aerosol algorithms discussed in the previous sections, and summarized in Tab. 1. The differences between the different reprocessings are synthesized in Tab. 2
for all collocated OMI-MODIS aerosol scenes.



### 4.1 Implicit aerosol correction - Benefits of the updated OMI cloud algorithm

Among all the main changes that are included in the updated version of OMCLDO2, the increased sampling of the OMI cloud LUT is expected to be the most important for the aerosol correction (see Sect. 2.2). Indeed, the coarse sampling of the OMI cloud LUT in the former version was clearly identified by Chimot et al. (2016) as a limitation regarding the behaviour and the magnitude of $c_p$ and thus, when deriving $N^v_{NO_2}$ in presence of aerosols (see Sect. 3.3). As depicted by Fig. 1, differences in $cp$ are now quite significant at low $\tau$. On average, $c_p$ values from OMCLDO2-New are lower of about 200 hPa than those from OMCLDO2-Old (with large standard deviation) over scenes with MODIS $\tau$(550 nm) $\leq$ 0.5. Indeed, low aerosol load has very limited effects on $N^s_{O_2-O_2}$ and does not dominate the measured radiance signal. This results in large uncertainties in the retrieved $c_p$ and a large sensitivity of the resolution at which the LUT interpolation is performed for these cases. Over scenes with high aerosol load (MODIS $\tau$(550 nm) $\geq$ 1.0), differences are more minor and may even reverse sign. We attribute the small reverse sign to the application of the temperature correction on $N^s_{O_2-O_2}$ (see Sect. 2.1) which, depending on the temperature difference compared to the assumed mid-latitude summer atmosphere, may apply a positive or negative small modification of $c_p$ in case of high $\tau$. However, as analysed by Veefkind et al. (2016), the impact of the temperature correction on $cp$ remains minor in case of high $cf$ and thus aerosol load, compared to the updated OMI cloud LUT. Overall, all these changes are consistent with those analyzed by Veefkind et al. (2016) over cloudy scenes, with low and high $c_f$.

Based on synthetic cases, Figure 2 illustrates the expected improvements of the implicit aerosol correction on $N^v_{NO_2}$ due to the higher OMI cloud LUT sampling. While remaining $N^v_{NO_2}$ biases were contained between $-20$ and $-40$ (%) with OMCLDO2-Old, they should be now limited to the range of [0:20]% with the use of OMCLDO2-New over scenes with relatively scattering aerosol particles (i.e. $\omega_0$=0.95) and assuming a typical NO$_2$ summer vertical profile over north-east China. Such improvements are particularly good in case of aerosols located at elevated altitude (i.e. more than 1 km). However, although improved, biases can be higher in case of more absorbing particles: i.e. in the range of $[-10:20]$ (%). Additional geophysical parameters, in particular the NO$_2$ profile shape, may affect these biases and are therefore of high importance (see further discussions in Sect. 5.2).

Overall, the future changes when applying the new implicit aerosol corrections from DOMINO will result from a combination of different parameters, mainly the higher sampling of the OMI cloud LUT, and then the temperature correction on $N^s_{O_2-O_2}$ and the updated OMI surface albedo database. To quantify the resulting changes in the reprocessed OMI $N^v_{NO_2}$, the results are separated in two steps. Firstly, Figures 3a, c and e illustrate the changes in reprocessed $N^v_{NO_2}$ from DOMINO to OMCLDO2-Old. As indicated in Tab. 1, these changes result from two consequences: 1) the temperature correction on $N^s_{O_2-O_2}$ and the new OMI surface albedo, which both directly modify the retrieval of the effective cloud parameters, and 2) the direct application of this new albedo when computing $A^v_{NO_2}$. A higher surface albedo should result in an increased length of the average light path, and therefore an enhanced $A^v_{NO_2}$. However, this can become more complex when combined with the new effective cloud parameters as they may either enhance, attenuate or even counterbalance this effect. On average, $N^v_{NO_2}$ is lower (i.e. higher $A^v_{NO_2}$), between -1 $\pm$ 9 (%) in China summertime and -15.6 $\pm$ 29.8 (%) in China wintertime. The quality of these

(c) Author(s) 2018. CC BY 4.0 License.





changes however depends on the accuracy of the new surface albedo climatology, which is primarily expected to be more robust due to the longer time series considered in the OMI reflectance observations (see Sect. 3.3).

Secondly, Figures 3b, d and f) depict the impacts of the the implicit aerosol correction evolution from OMCLDO2-Old to OMCLDO2-New. They are directly driven by the improved $c_p$ (see Tab. 1). Over scenes with MODIS $\tau$(550 nm) in the range

of [0.0:0.5], a decreased $c_p$ (see Fig. 1) results in a stronger shielding (or reduced enhancement) effect from particles: $N_{NO_2}^v$ generally increases. In contrast, larger $c_p$ over scenes with MODIS $\tau$(550 nm) $\geq 1.0$ leads to a lower shielding (or stronger enhancement) effect: $N_{NO_2}^v$ decreases. Standard deviation of these changes is between 15 and 20 % in China wintertime and South America, and lower than 10% in China summertime and South America. Averages are in the range of [1.3:7.9]%. Regional and seasonal differences may reflect the implicit dependencies on the aerosol types, the combined effects on $c_f$-$c_p$

and the impacts of seasonal NO$_2$ vertical profile. All these observed changes are in line with the analyses deduced from the synthetic cases in Fig. 3b, d and f, confirming the improvements thanks to the updated OMI cloud LUT. Interestingly, these overall changes seem to be in line with the average AMF uncertainty of 11% evaluated by Lorente et al. (2017) due to different cloud correction scheme in polluted conditions and assuming $c_f \leq 0.2$.

Overall, maps in Fig. 4c-6c show that the total changes in $N_{NO_2}^v$, from DOMINO to OMCLDO2-New, mostly occur in

the eastern part of China, where the NO$_2$ pollution is higher. Spatial patterns of these overall changes mostly result from a complex combination with MODIS aerosol horizontal distribution as suggested by Fig. 3, but also aerosol types, and vertical distribution: a decrease over Bejing areas in summertime, and an increase in the same area in wintertime.

## 4.2   Explicit aerosol correction results

In this study, there are four possibilities to apply an explicit aerosol correction from the OMI 477 nm O$_2$-O$_2$ band. Each of

them differ regarding the assumed aerosol properties (i.e. $\omega_0$), aerosol $\tau$ (i.e. MODIS or OMI), and the consequent fitted ALH (see. Sect. 2.3). All these possibilities were overall applied when reprocessing the OMI DOMINO product to quantify their overall differences.

Similarly to the benefits of the new implicit aerosol correction based on OMCLDO2-New evaluated in Sect. 4.1, Fig. 7 shows the benefits of the applied explicit aerosol corrections. Provided that the aerosol model (e.g. $\omega_0$) is in line with the actual

aerosol type present in the observed scene (i.e. ideal scenario), remaining biases on $N_{NO_2}^v$ are below 20%, and little dependent on aerosol parameters ($\tau$, $\omega_0$ and ALH) when assuming a NO$_2$ vertical profile representative of a typical day in summertime over China east (van Noije et al., 2014; Chimot et al., 2016). In such a scenario, using either the retrieved OMI $\tau$ value or a more accurate one is not expected to make a major difference. However, in practice, these results may vary with respect to the NO$_2$ profile shape and additional errors in the employed aerosol model (see next subsections).

Fig. 8 shows that all reprocessed $N_{NO_2}^v$ with implicit or explicit corrections are larger by [10:50]% than if no correction was performed, especially over scenes with MODIS $\tau$(550 nm) $\geq 1.0$. This suggests that both strategies converge to the same direction (i.e. same sign) in spite of some different magnitudes of the aerosol correction. Since all the considered strategies attenuate the $N_{NO_2}^v$ biases due to aerosols from an aerosol-free scene assumption, it is worth emphasizing that all of them, without distinction, are an aerosol correction, regardless of their implicit or (more) explicit nature.



Overall, over OMI pixels collocated with MODIS $\tau(550\,nm) \geq 0.5$, Fig. 9 depicts that most of the reprocessed $N_{NO_2}^v$ values are generally higher with the explicit aerosol correction than with the implicit aerosol correction from OMCLDO2-New. This then generally suggests a stronger shielding effect generally applied by these aerosol correction strategies leading to lower $A_{NO_2}^v(439\,nm)$. The differences increase with increasing MODIS $\tau$ as aerosol effects consequently amplify along the average light path.

In eastern China, by using the explicit aerosol correction with $NN_{MODIS,\omega_0=0.95}$, $N_{NO_2}^v$ values are higher than with the implicit aerosol correction with OMCLDO2-New by about $12 \pm 12.5$ % in summer and $40 \pm 26.1$ % in winter over scenes with MODIS $\tau(550\,nm) \geq 0.5$ (see Fig. 9). The larger increase in wintertime is likely due to different $NO_2$ profiles, with $NO_2$ molecules being closer to the surface (see further discussions in Sect. 5.2). The differences with OMCLDO2-New are somehow reduced when assuming a lower aerosol $\omega_0$. In such a configuration, the main differences are: 1) a lower ALH due to an assumed lower $\omega_0$ value (Chimot et al., 2017), combined with 2) a more absorbing aerosol model used to compute $A_{NO_2}(439\,nm)$. In both cases, $NN_{MODIS,\omega_0=0.9}$ and $NN_{MODIS,\omega_0=0.95}$, prior $\tau$ (coming from MODIS) remains unchanged. As illustrated in Figs. 9, resulting $N_{NO_2}^v$ using $NN_{MODIS,\omega_0=0.9}$ over China are smaller (i.e. $A_{NO_2}^v(439\,nm)$ higher) by about $-0.2\% \pm 7.8\%$ in summer, and $-8.2\% \pm 22.3\%$ in winter compared to $N_{NO_2}^v$ with $NN_{MODIS,\omega_0=0.95}$. These numbers represent then a first evaluation of the impact of aerosol model uncertainty, assuming one may use a very accurate prior $\tau$ information for both ALH retrieval and then $A_{NO_2}^v(439\,nm)$ computation.

Over scenes in South America with MODIS $\tau(550\,nm) \geq 0.5$, the difference between $N_{NO_2}^v$ from $NN_{MODIS,\omega_0=0.95}$ and from OMCLDO2-New is on average close to zero with a standard deviation of 16.8%. The use of $NN_{MODIS,\omega_0=0.9}$ reduces $N_{NO_2}^v$ by about $-1.3\% \pm 8.7\%$. Interestingly, Castellanos et al. (2015) reported an average change of $0.6\% \pm 8\%$ on $A_{NO_2}^v$ after reprocessing DOMINO $N_{NO_2}^v$ over cloud-free scenes during the biomass burning season in South America and applying an explicit aerosol correction based on the OMI near-UV aerosol algorithm (OMAERUV) and CALIOP aerosol ALH.

When using the retrieved OMI $\tau$ as prior input instead of MODIS $\tau$ over eastern China, $N_{NO_2}^v$ differences with respect to the use of OMCLDO2-New differ by 5-10% on average over scenes with MODIS $\tau(550\,nm) \geq 0.6$ (higher in summer but lower in winter). This suggests then a higher sensitivity to the combination of OMI $\tau$ and ALH when used together for the $A_{NO_2}^v$ computation.

Figs. 4-6 show that most of the changes in $N_{NO_2}^v$ are located on the eastern part, and over areas dominated by heavy $NO_2$ pollutions such as the mega-cities and the Pearl River Delta. Horizontal distribution of aerosol load adds some complex patterns.

Overall, the quantitative $N_{NO_2}^v$ differences between the applied explicit aerosol corrections and the improved implicit aerosol correction can be considered as an average uncertainty related to the choice of an aerosol correction approach. Similar numbers are reported by Lorente et al. (2017) who indicated an average aerosol correction uncertainty of 45% over highly polluted scenes and with large aerosol loading ($\tau(550\,nm) \geq 0.5$). Furthermore, it was found that $N_{NO_2}^v$ from POMINO dataset over China (Lin et al., 2015) is 55% higher ($A_{NO_2}$ smaller) than if no explicit aerosol correction was considered when aerosol layer is located above the tropospheric $NO_2$ bulk. The main identified reason was a lower screening applied when only the effective cloud parameters were used as the Lambertian reflector was defined at a lower altitude ($c_p = 350$ hPa). This may of course be



attenuated when aerosols are mixed with $NO_2$. as their multiple scattering effects increase then the average light path length and thus the $NO_2$ absorption.

Finally, over scenes with little aerosol amount (i.e. MODIS $\tau$(550 nm) $\leq 0.2$), the difference in $N_{NO_2}^v$ between the explicit aerosol correction assuming prior MODIS $\tau$ and the implicit aerosol correction with OMCLDO2-New is systematically lower and non-null: about an average of -10% over all the considered regions. Such a difference may seem strange as so little aerosol amounts are expected to have an almost negligible effect on the light path and thus on $A_{NO_2}^v$. When OMI $\tau$ is instead considered, this difference becomes positive over China, but is reduced in absolute everywhere (less than 10%). Note that $NN_{MODIS,\omega_0=0.95}$ and $NN_{MODIS,\omega_0=0.9}$ algorithms differ from OMCLDO2 by using an external geophysical parameter (i.e. MODIS $\tau$). Although more accurate than using the retrieved OMI $\tau$, the combination of an external MODIS aerosol paremeter derived from different assumptions about the scattering model and surface reflectance may at the end lead to inconsistencies when combined with the OMI NN model: the 477 nm $O_2 - O_2$ radiance budget is likely not closed. This radiance budget is always closed with OMCLDO2 (apart from the instrument noise) since it simultaneously adjusts both $cf$ and $cp$ to match with the $R_c$(475 nm)-$N_{O_2-O_2}^s$ combination. The topic of radiance closure budget and its impacts on $A_{NO_2}^v$ are further discussed in Sect. 5.5.

## 4.3 Explicit vs. implicit aerosol correction - Main reasons of the differences

As discussed in Sect. 3.2, ALH is the first crucial parameter for the computation of $A_{NO_2}^v$(439 nm). Therefore, as a first assumption, it is expected that the accuracy of the OMI ALH retrieval, and its difference with $c_p$, may be one of the first causes (although not unique) of the difference between the applied implicit and explicit aerosol correction.

Figure 10 compares the average OMI ALH (retrieved with the NN trained with aerosol $\omega_0 = 0.95$) and $c_p$ from OMCLDO2-New, both converted in metric unit (km) over cloud-free scenes and for collocated MODIS scenes with $\tau$(550 nm) $\geq 0.5$. Overall, both variables are quite well correlated, with similar spatial and seasonal distributions. Values are higher in China summertime and over South America and lie in the range of [1.5:5.0] km. They are lower in China wintertime, between 0.4 and 2.0 km. Quantitatively, ALH values from $NN_{MODIS,\omega_0=0.95}$ show that the fitted aerosol layers are located higher than the fitted opaque Lambertian clouds: i.e. aerosol pressure are smaller than $cp$ with average differences in the range of $[-0.49:-50.3]$ hPa. Standard deviation of the differences are in the order of 120 hPa. The sign of the differences is reversed when employing $NN_{MODIS,\omega_0=0.9}$ (average differences 12.9-59.3 hPa).

As a first assumption, when ALH values are higher than $c_p$, the explicit aerosol corrections shall generally apply a stronger shielding effect on $A_{NO_2}^v$. Therefore, the resulting $N_{NO_2}^v$ should be larger. However, this only element is likely insufficient to explain the differences observed in Sect. 4.2. The combined impacts of the assumed prior $\tau$ value also play a significant rule. Furthermore, the assumed Lambertian cloud and the aerosol Henyey-Greenstein models differ about the horizontal coverage of the OMI pixel: the opaque Lambertian cloud model only covers part of the pixel (the fraction coverage is fitted through $c_f$, optical properties are fixed). The clear pixel fraction ensures the transmission part of the signal and the related multiple scattering not present by definition within the Lambertian cloud layer. In contrast, the aerosol model (and analysed synthetic cases) cover the whole pixels (optical properties can be changed, fraction coverage is fixed). The transmission and multiple





scattering properties are included within the aerosol layer and vary as a function of the optical properties. Therefore, one can assume that in case of optically thick layers, the aerosol model generally applies a stronger screening effect by fully covering the scene and thus obstructing the surface transmission signal. By opposition, the surface transmission signal is more or less always ensured with the Lambertian opaque model by the non-covered pixel fraction.

## 5 Advantages and challenges of an explicit aerosol correction based on the 477 nm $O_2$-$O_2$ measurement

In this section, we want to discuss specific elements to evaluate the relevancy of the developed explicit aerosol correction strategy over cloud-free scenes from OMI, but also in general from all UV-vis satellite measurements. In particular, we wish to draw the reader's attention to the advantage of using an explicit aerosol correction based on the exploitation of the 477 nm $O_2$-$O_2$ spectral band, but also the remaining difficulties to implement it in practice. The next subsections focus on the significance of the aerosol model error, the importance of the $NO_2$ vertical profile, the cases with absorbing particles, the $NO_2$ vertical averaging kernels, and the OMI visible radiance closure budget issue.

### 5.1 Impact of aerosol model error on tropospheric $NO_2$ air mass factor

When applying an explicit aerosol correction, the accuracy of each variable describing aerosol properties, once combined with the $NO_2$ vertical profile (see Sect. 5.2) and surface reflectance, overall drives the $A_{NO_2}^v$(439 nm) computation uncertainty. The $A_{NO_2}^v$ uncertainty due to this whole set of variables, not only ALH alone, can be defined as the aerosol model error for OMI $N_{NO_2}^v$ retrieval.

To understand, the quantitative impact of each aerosol model variable uncertainty, Figure 11 shows the $A_{NO_2}^v$(439 nm) biases resulting from some aerosol model inputs. A single bias on ALH of 100 hPa directly affects $N_{NO_2}^v$ within the range of [60:70]% when absorbing aerosols ($\omega_0 = 0.9$) are located below 0.5 km, assuming wintertime $NO_2$ profile and with $\tau$(550 nm) = 1.4. The uncertainties are below 50% for $\tau$(550 nm) $\leq 0.5$, and overall below 10% when particles are located at elevated altitudes (i.e. true ALH $\geq 1.4$ km). This quantitatively emphasizes how essential is ALH information quality when particles are actually mixed with $NO_2$ molecules due to the complexity to reproduce the enhancement of the average light path caused by scattering effects. A bias of 0.2 on the assumed $\tau$(550 nm) mostly impacts scenes with little aerosol load: while resulting $N_{NO_2}^v$ uncertainties lie in the range of [−20:20]% for $\tau$(550 nm) $\leq 0.5$, they decrease to the range of [0:10]% for $\tau$(550 nm) = 1.4. Finally, an overestimation of aerosol scattering efficiency (i.e. $\omega_0$ bias of 0.05) leads to an underestimation of $N_{NO_2}^v$ up to −20% over scenes with high $\tau$ as a consequence of an underestimation of aerosol shielding effect and therefore a too large $A_{NO_2}^v$(439 nm). Overall, ALH uncertainty is the key driver of the AMF computation quality. It requires to be better than 50 hPa to limit $N_{NO_2}^v$ bias below 40%. With $\tau$ uncertainty, they form the most important set of aerosol parameters prerequisite for for a high quality of $A_{NO_2}^v$(439 nm) computation. Although not negligible, uncertainty of aerosol model parameters that are more related to the particle optical and scattering properties, such as $\omega_0$, $g$ and $\alpha$, is of second importance provided that both ALH and $\tau$ qualities are ensured.



## 5.2  The importance of the relative layer height

A comprehensive aerosol correction for an accurate $A_{NO_2}$(439 nm) computation also requires the actual $NO_2$ vertical profile. Figure 12a shows the accuracy of the aerosol corrections in $A_{NO_2}$(439 nm), based on synthetic case, assuming the presence of absorbing aerosol particles ($\omega$=0.9) but a $NO_2$ vertical profile of winter time (1st of January, 12:00 pm) over China. The main difference with Fig. 2 and Fig. 7 is the presence of a more abundant tropospheric $NO_2$ bulk closer to the surface, and a stronger decrease rate to higher altitudes (Chimot et al., 2016). In such a case, relative $N^v_{NO_2}$ biases with the implicit aerosol correction are strongly degraded from $[-10:20]\%$ (summertime) to $[-80:40]\%$ (winter). Already identified in Sect. 4.1 with the summer time $NO_2$ profile, the insufficient shielding effect applied by the effective cloud parameters from OMCLDO2-New in case of aerosol layers located in elevated altitude is here severely degraded (from $-10\%$ to $-80\%$). The insufficient enhancement effect when particles are mixed with the tropospheric $NO_2$ molecules is also here amplified (from 20% to 40%).

When considering an explicit aerosol correction using $NN_{MODIS,\omega_0=0.9}$, $N^v_{NO_2}$ bias is changed to 0-40 (%). Similarly to summertime, they are lower in case of particles at high altitude, suggesting then the strong benefits of such a correction scheme in wintertime and/or in presence of absorbing particles. The cases of aerosols close to the surface (i.e. lower than 0.5 km) remains an issue due to the difficulty to distinguish the scattering effects from the surface and the adjacent aerosol layer when retrieving ALH. The retrieval seems, in such a case, to overestimate the aerosol layer altitude.

The observed positive difference of about 40% between explicit and implicit aerosol corrections is in line with the analyses over China in wintertime (see Sect. 4.2 and Fig. 9). All these elements strongly remind and emphasize that the quality of the aerosol corrections, and their differences, for $N^v_{NO_2}$ retrieval is actually more dependent on the relative height between the particles and the tropospheric $NO_2$ bulk, than ALH or $cp$ themselves. The evolutions of operational aerosol correction schemes in present and future air quality UV-visible space-borne sensors, such as TROPOMI on-board Sentinel-5 Precursor, need then to consider a proper joint characterization of trace gas vertical profiles together with aerosol vertical distribution and related optical properties.

## 5.3  Explicit vs. implicit aerosol correction - Focus on absorbing particles

In this study, the cases with absorbing particles, i.e. aerosol $\omega_0 = 0.9$, brings more challenges to the implicit aerosol correction than scenes dominated by more scattering aerosols. Such difficulties were also identified by Castellanos et al. (2015). For reminder, the presence of absorbing particles leads to a reduced $c_p$ (increasing the shielding effect) but a lower $c_f$ (lower shielding effect) with then a higher transmittance coming from the clear part of the pixel. Part of the insufficient shielding effect observed with the implicit correction in Sect. 4.1 and 5.2 may thus be explained by an insufficient coverage of the observation scene.. This last element likely overall limits the potential of the effective cloud model to apply an adequate shielding effect on $A^v_{NO_2}$. However, these biases still remain lower than if no aerosol correction was achieved (Chimot et al., 2016).

Figures 7 and 12a showed the potentiel of the explicit aerosol correction, but by assuming no bias in the assumed $\tau$ and $\omega_0$ parameters. To investigate further, Fig. 12c)-d) depict the impacts of a wrong prior aerosol $\omega_0$ (overestimation of 0.05) when applying the explicit aerosol correction through the use of the OMI aerosol NN. When the true prior $\tau$ value for both ALH



retrieval and $A_{\mathrm{NO_2}}^{\mathrm{v}}$ computation is conserved the increases of $N_{\mathrm{NO_2}}^{\mathrm{v}}$ biases are limited: from 0-40% (cf. Fig. 12a) to 0-60%. This reflects how the quality of prior $\tau$ knowledge helps to constrain both ALH retrieval, and then limit the perturbation on $A_{\mathrm{NO_2}}^{\mathrm{v}}$. However, this may be more severely degraded when inaccurate prior $\tau$ is considered (see Sect. 5.5). Figure 12d) shows the impacts of replacing true $\tau$ by the retrieved OMI one. $N_{\mathrm{NO_2}}^{\mathrm{v}}$ degradations are more important, about $[-40:60]$%. This is

a direct result of a degradation of ALH retrieval due to a too much biased aerosol $\tau$ and the resulting impact on the $A_{\mathrm{NO_2}}^{\mathrm{v}}$. As discussed in Sect. 5.1, although $\omega_0$ parameter is of second importance for $A_{\mathrm{NO_2}}^{\mathrm{v}}$(439 nm) itself compared to the (ALH-$\tau$) combination, its direct impact on $\tau$ retrieval consequently affects then ALH determination accuracy and then indirectly $A_{\mathrm{NO_2}}^{\mathrm{v}}$.

     In spite of these drawbacks, $N_{\mathrm{NO_2}}^{\mathrm{v}}$ biases remain smaller than if the derived effective cloud parameters are employed through the implicit aerosol correction (cf. Fig. 12a). Thus, even if imperfect, the explicit aerosol correction based on OMI ALH and $\tau$

retrievals seem to remain still advantageous for an efficient aerosol correction in tropospheric NO2 VCD retrieval from visible satellite radiance. Nevertheless, true horizontal distributions of aerosols within the observation pixel may actually be quite heterogeneous. Such problems should be further investigated with a focus on areas where absorbing particles are expected such as in winter time in China, or during large biomass burning episodes. Future studies should also determine how often such effect occurs and its overall impact depending on the NO2 vertical profile variability.

## 5.4    Aerosol model and NO2 vertical averaging kernel

Theoretically, the application of an explicit aerosol model is expected to simulate more realistic scattering and absorption effects due to particles when computing $A_{\mathrm{NO_2}}$(439 nm). Figure 13 illustrates the vertical averaging kernel (AK) (cf. Sect. 3.2) assuming $\tau$(500 nm) = 1.0 and from the application of OMCLDO2-New, $NN_{MODIS,\omega_0=0.95}$ and true aerosol conditions, assuming no bias in prior aerosol assumptions. AKs are very important for estimating the surface $NO_x$ emissions by convoluting

the NO2 vertical profiles from the used atmospheric models to match with the OMI $N_{\mathrm{NO_2}}^{\mathrm{v}}$ observations (Eskes and Boersma, 2003; Ding et al., 2015). AKs based on OMCLDO2-New display a sharp distinction between the enhanced atmospheric layers located above aerosols and the shielded layers located below. On the contrary, AKs from $NN_{MODIS,\omega_0=0.95}$ depict a smoother transition from then enhanced to the shielded layers. This transition is more in line with the actual AKs (Fig. 13a) and results from the scattering effects induced by the more or less wide aerosol particle layer. A bright Lambertian reflector is by nature

fully opaque and does not induce multiple scattering effects. This is partly compensated by the transmission of the clear fraction of the pixel through the IPA assumption. This suggests that applying an explicit aerosol correction leads to the consideration of more realistic physical assumptions and AK productions. However, such a suggestion critically depends on an ensemble of other parameters that contribute to the AK generation: the accuracy of the retrieved ALH which triggers the location of the enhancement / shielding transition, the potential aerosol model biases (e.g. $\omega_0$, scattering phase function, etc...), and the

difference between actual and assumed aerosol vertical profile.

## 5.5    Radiance closure budget issue and potential impacts

The discussions about the "model error" in the previous sub-sections implicitly made the assumption that the whole set of parameters such as ($cf$-$cp$-surface reflectance) on the one hand, or ($\tau$-ALH-$\omega_o$-$g$-$\alpha$-surface reflectance) are fully consistent in

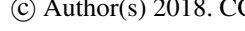


the sense that they form one unique particle model. However, when external data are used to constrain ALH retrieval accuracy, such as MODIS aerosol $\tau$, one may combine inconsistent model assumptions together leading then to complex artefacts such as the issue of OMI closure radiance budget.

Radiance closure budget is not only important in the 477 nm $O_2$-$O_2$ band, but also in the $NO_2$ absorption band at the wavelength where the AMF is computed. Aerosol $\tau$ combined with surface reflectance are expected to drive OMI $R_c$(475 nm). As discussed in Sect. 2.2, OMCLDO2 simultaneously adjusts both $cf$ and $cp$ based on the same prior surface reflectance such that their combination allows to close the ($R_c$(475 nm)-$N^s_{O_2-O_2}$) budget, and thus the OMI 477 nm $O_2 - O_2$ radiance regardless the accuracy of the selected model. On the contrary, by using MODIS $\tau$, only OMI $N^s_{O_2-O_2}$ is exploited, not $R_c$(475 nm). If all the prior parameters are accurate and derived from a unique set of aerosol model and surface reflectance parameters, the ($R_c$(475 nm)-$N^s_{O_2-O_2}$) budget should be closed. However, any mismatch between the model employed for MODIS $\tau$ determination and the one used in the OMI NN training dataset, between MODIS and OMI instrument radiance, and/ or the surface reflectance hypothesis may leave this budget open. In particularly, it is worth reminding that surface reflectance dataset behind MODIS $\tau$, OMI $cf$ and $A^v_{NO_2}$(439 nm) are not identical: for OMI , a multi-spectral surface Lambertian equivalent reflectance (LER) was used(Kleipool et al., 2008) while MODIS aerosol retrieval uses a directional surface spectral reflectance (Levy et al., 2013).

Therefore, the strategies of implicit vs. explicit aerosol correction analysed in Sect. 4.2 do not only differ in terms of assumed particle optical and scattering property model but also on how much the whole OMI radiance budget is eventually fitted. This last difference likely explains the strange systematic difference (about -10%) identified over OMI scenes with MODIS $\tau$(550 nm) $\leq 0.2$, where aerosol effects could be assumed as almost insignificant (see Fig. 9). When. MODIS $\tau$ is replaced by the retrieved OMI $\tau$ (see. Tab. 1) which is, like $cf$, mostly constrained by $R_c$(475 nm), prior OMI surface albedo and the same aerosol model used for ALH retrieval and $A^v_{NO_2}$(439 nm) calculation, resulting $N^v_{NO_2}$ are higher than with the use of MODIS $\tau$ (see Tab. 2). Moreover, as analysed in Sect. 4.2, the differences between the implicit and explicit aerosol corrections are in absolute smaller everywhere over scenes with small aerosol load confirming the consistency in the employed models for each set of parameters and an almost complete closure of the OMI visible radiance budget (instrument noise apart).

At the end, one might wonder what the best option is for an optimal aerosol correction: 1) using the best aerosol and surface parameters available for the most accurate correction at the cost of not closing the satellite radiance budget, or 2) applying a less accurate correction but with an ensemble of aerosol and surface parameters that eventually comprehensively fit the spectral measurement. The first option gives more weight to the used auxiliary data, while the second option maximises the weights of the amount of information contained in the satellite measurement. The answer to such a problem is, in our opinion, not clear at this moment. But, given the fact that several studies prioritise the application of multiple parameters from very diverse sources (models, ancillary instruments with different techniques, etc..) to satellite spectral measurements, we think that the issue of radiance closure budget should be kept in mind by the scientific community and further investigated in future research studies. At the end, an optimal trade off must be found between quality of $N^v_{NO_2}$ product and the weights given to the original satellite measurement.



## 6 Conclusions

This paper analyses the reprocessing of the reference OMI tropospheric $NO_2$ vertical column density $N_{NO_2}^v$ for cloud-free scenes over east China and South America for 2006 and 2007. These regions are dominated by high aerosol loadings. The goal of this study is to evaluate the benefits of the recent achieved developments during the last years to improve the so-called aerosol correction in the tropospheric $NO_2$ air mass factor $A_{NO_2}^v$, a crucial parameter when retrieving $N_{NO_2}^v$ from visible air quality satellite backscattered measurements. In particular, the tested aerosol correction strategies rely on our recent experiences with the 477 nm $O_2$-$O_2$ spectral band: a key spectral band present in current and future air quality UV-vis satellite instruments such as OMI, GOME-2, TROPOMI, Sentinel-4-UVN, and Sentinel-5-UVNS. The use of this band is important in view of operational data processing as it allows to derive parameters that, in principle, could reproduce the particle scattering effects on the average light path in the visible spectral window and representative of the sensor field of view. Regardless of the type of algorithm employed here, these parameters can all be derived using computationally inexpensive methods, are directly representative of the OMI pixel in terms of spectral measurement, spatial coverage and temporal acquisition, and can directly be used for the $A_{NO_2}^v$ computation.

The two tested algorithms are the OMI cloud software OMCLDO2 (Veefkind et al., 2016), and the OMI aerosol Neural Network (NN) approach (Chimot et al., 2017, 2018). The most important difference between these methods is the assumed aerosol model, not only for the retrieval of the aerosol parameters in the $O_2$-$O_2$ band, but also how they represent the aerosol effect in the tropospheric $NO_2$ air mass factor. The OMCLDO2 represents the aerosols as effective cloud parameters and thus implicitly correct the aerosol effects, whereas the NN approach explicitly models and corrects for aerosols. The most recent OMCLDO2 update from (Veefkind et al., 2016) includes, among many elements, an increased number of nodes in the lookup-tables and the necessary temperature correction on the $O_2$-$O_2$ slant column density.

For both methods, the reprocessed $N_{NO_2}^v$ shows smaller biases over cloud-free scenes dominated by aerosol pollution, compared to the standard DOMINO retrieval and its related aerosol correction. Previous studies showed an underestimation of the OMI $N_{NO_2}^v$ between $-20\%$ and $-40\%$ over scenes with MODIS aerosol optical thickness $NO_2$ $\tau(550 \text{ nm}) \geq 0.6$, assuming scattering aerosol particles with aerosol single scattering albedo $\omega_0 = 0.95$ and summertime $NO_2$ vertical profile. In similar conditions, these biases are expected to be contained in the limit of $[0:20]\%$, for both the most recent OMCLDO2 method and the NN method. On average, the applied explicit NN correction leads to higher $N_{NO_2}^v$ values compared to the implicit OMCLDO2 correction up to 40% depending on the seasons, regions and aerosol pollution episodes. This represents our best estimate of the aerosol correction uncertainty for the OMI $N_{NO_2}^v$ retrieval due to the possible choices of the algorithm correction. They are attributed to the model differences and the associated output variables.

Although both the implicit OMCLDO2 and explicit NN methods are a significant improvement over the current OMI DOMINO retrieval, we also found limitations that need further study:

- If absorbing aerosols (i.e. single scattering albedo $\omega_0 \leq 0.9$) are present, the implicit aerosol correction still leads to substantial biases on $N_{NO_2}^v$ due to an insufficient applied shielding effect: between $-80\%$ and $20\%$ assuming a typical $NO_2$ wintertime profile . This is most severe when particles are located at high altitude (above 1.4 km) and with heavy




aerosol load. In similar conditions, the explicit aerosol correction allows to mitigate the biases to the range of [0:40]% if accurate $\tau$ and aerosol model are available, or [?40:60]% with inaccurate $\tau$ (e.g. retrieved OMI $\tau$) and aerosol model.

– Biases remain high when particles are located close to the surface, regardless of the aerosol correction methodology. The distinction between aerosol scattering and surface reflectance is challenging under such conditions. Since particles are close to the tropospheric $NO_2$ bulk (either mixed or slightly above), small errors in the retrieved aerosol height or effective cloud pressure have a significant impact on the derived $NO_2$ column.

– All aerosol correction methodologies are sensitive to the $NO_2$ vertical profile and the employed surface albedo or reflectance. Future improvements need to address both these parameters together with any evolution that can still be done in the aerosol or effective cloud retrieval from the 477 nm $O_2$-$O_2$.

– Although the OMI aerosol NN algorithms lead to promising ALH retrieval, its use for an explicit aerosol correction is not straightforward. Indeed, ALH is only one variable (although a key one) of a set of aerosol parameters that need to be applied for the required explicit aerosol correction. Its combination with other aerosol parameters (size, $\omega_0$, $\tau$) can lead either to model errors and/or to the risk of not closing the OMI radiance budget if any of these parameters was issued from an external source (model or instrument) not consistent with the OMI ALH retrieval.

Based on its estimated performances, we recommend for an operational processing of OMI data, where no distinction is made between cloudy, cloud-free, and aerosol contaminated scenes, to first use the implicit aerosol correction based on OMCLDO2. It allows to correct adequately both cloud and aerosol particle effects on the average light path. If, cloud-free scenes can be carefully identified and collected in a same way it has been done here, then the explicit aerosol correction based on the OMI NN aerosol algorithms should be considered. We demonstrated in this study, for the first time, its high performance and the realism of the simulated physical effects. Moreover, the developed NN approach can ensure a fast cloud-free data processing.

Overall, the considered aerosol corrections can easily be transposed to the future generation of air quality UV-vis satellite sensors, such as TROPOMI, Sentinel-4, and Sentinel-5-UVNS. It can also be considered for other trace gases of interest: e.g. $O_2$, HCHO. However, such an approach has to be adapted to the specificities of this new generation of instruments.

*Code and data availability.* All the data results and specific algorithms created in this study are available from the authors upon request. If you are interested to have access to them, please send a message to Julien.Chimot@eumetsat.int and pepijn.veefkind@knmi.nl. The pybrain library code is available at http://pybrain.org/. Finally, The OMCLDO2 dataset is available from the NASA archives: https://disc.gsfc.nasa.gov/uui/datasets/OMCLDO2_003/summary.

*Competing interests.* The authors declare that they have no conflict of interest.



*Acknowledgements.* This work was funded by the Netherlands Space Office (NSO) under the OMI contract.The authors thank Johanna Tamminen from the Finnish Meteorological Institue (FMI), Thomas Wagner from Max Plank Institute (MPI), Ilse Aben from SRON Netherlands Institute for Space Research and Folkert Boersma from KNMI for the reviews and discussions.





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




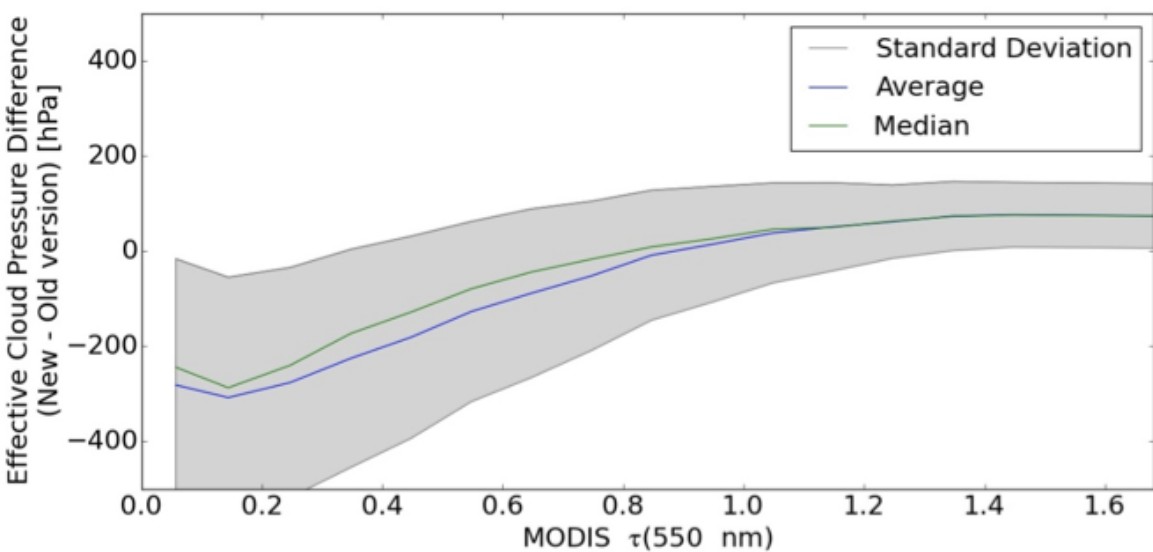

**Figure 1.** Statistics of effective cloud pressure differences between OMCLDO2-new and OMCLDO2-old (see Sect. 2.2 and Tab. 1) in [hPa] in 2006-2007 as a function of MODIS aerosol optical thickness (AOT) $\tau$(550 nm). An example over China in summertime (June-July-August).





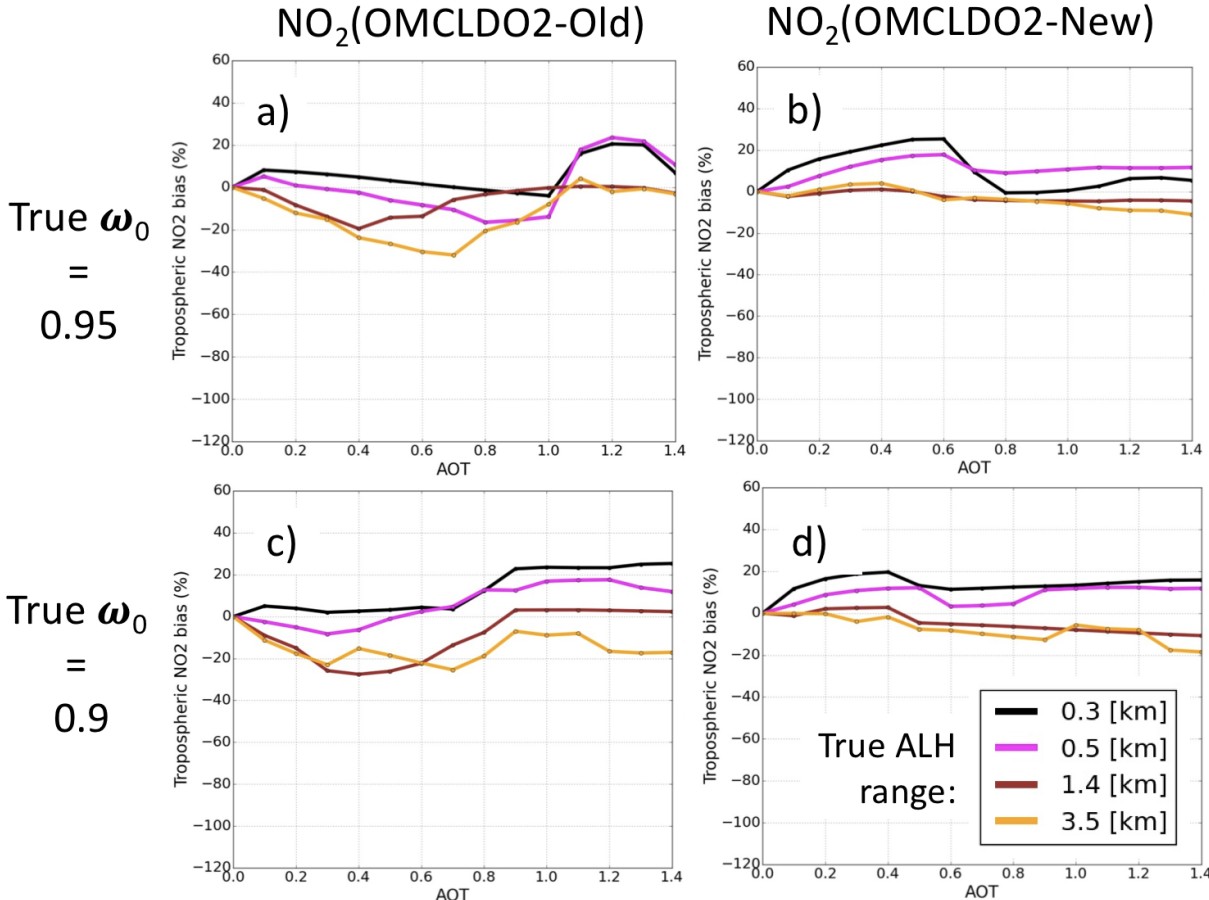

**Figure 2.** Relative $N_{NO_2}^v$ biases after application of the implicit aerosol correction as a function of true aerosol optical thickness (AOT) $\tau(550$ nm) and based on synthetic cases including different true ALH, surface albedo $= 0.05$, $\mu_0 = 25°$, $\mu = 25°$, and a typical TM5 $NO_2$ vertical profile for 1st of July, 2006 at 12:00 pm over China (van Noije et al., 2014; Chimot et al., 2016). True aerosol properties are defined by $\alpha = 1.5$, $\omega_0 = 0.95$ or $0.9$ and $g = 0.7$. Implicit aerosol correction is derived from the retrievals given by OMCLDO2-Old or OMCLDO2-New (see Sect. 2.2 and Tab. 1): **(a)** Relative $N_{NO_2}^v$ bias resulting from OMCLDO2-Old, true $\omega_0 = 0.95$, **(b)** Relative $N_{NO_2}^v$ resulting from OMCLDO2-New, true $\omega_0 = 0.95$, **(c)** Relative $N_{NO_2}^v$ bias resulting from OMCLDO2-Old, true $\omega_0 = 0.9$, **(d)** Relative $N_{NO_2}^v$ resulting from OMCLDO2-New, true $\omega_0 = 0.9$.

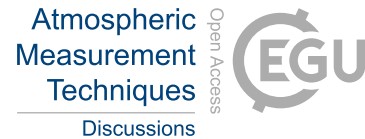

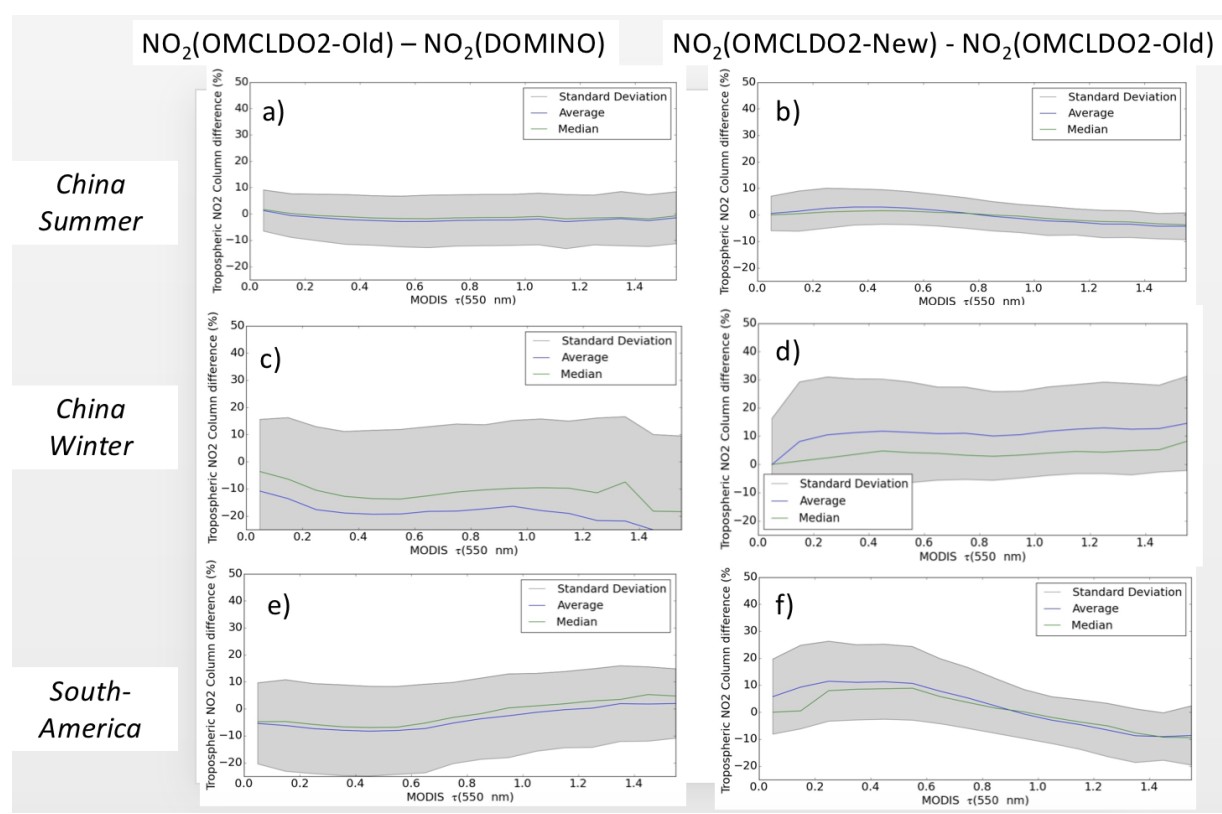

**Figure 3.** Statistics of relative $N_{NO_2}^v$ differences in (%) as a function of MODIS $\tau$(550 nm) over China and South America in 2006-2007 due to changes in the applied implicit aerosol correction (cf. Sect. 4.1) from DOMINO to OMCLDO2-Old, and from OMCLDO2-Old to OMCLDO2-New: **(a)** China Summer (June-July-August), **(b)** China Winter (December-January-February), **(c)** South America (August-September).



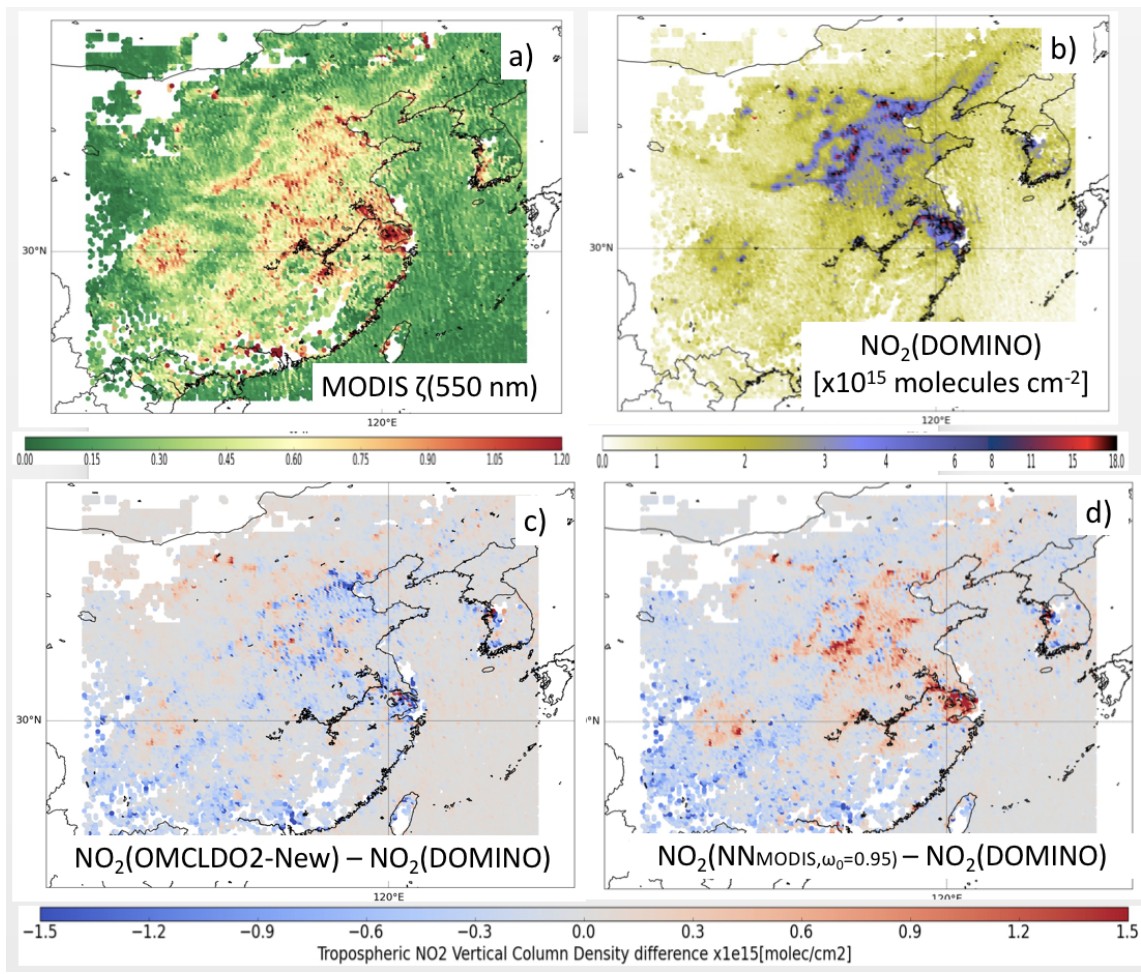

**Figure 4.** Average maps of MODIS $\tau$(550 nm), OMI DOMINO $N_{NO_2}^v$ and differences after applying the implicit (with OMCLDO2-New) or explicit (with $NN_{MODIS,\omega_0=0.95}$) aerosol correction over China in summertime (June-July-August) 2006-2007: **(a)** MODIS $\tau$(550 nm), **(b)** OMI DOMINO $N_{NO_2}^v$, **(c)** OMI $N_{NO_2}^v$ differences due to changes between OMCLDO2-New and DOMINO implicit aerosol corrections, **(d)** $N_{NO_2}^v$ differences between explicit aerosol correction based on the $NN_{MODIS,\omega_0=0.95}$ aerosol parameters (i.e. aerosol forward model assuming $\omega_0 = 0.95$, MODIS $\tau$(550 nm), and retrieved ALH) and implicit aerosol correction implemented in DOMINO.



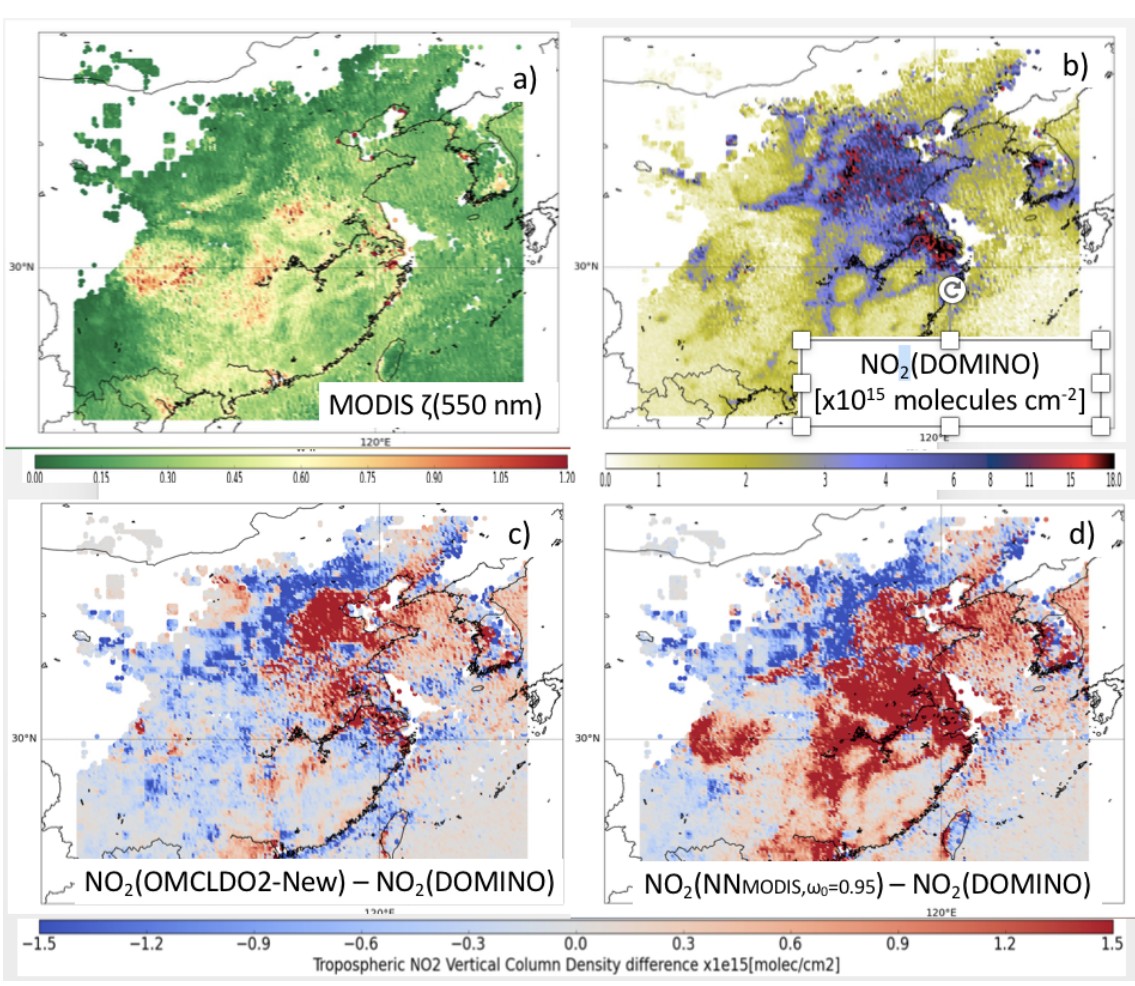

**Figure 5.** Same as Fig. 4 but over China in wintertime (December-January-February) 2006-2007





**Figure 6.** Same as Fig. 4 but over over South America during the biomass burning season (August-September) in 2006-2007




**True $\boldsymbol{\omega}_0$ = 0.95**

a) $NO_2(NN_{True}, \omega_0=0.95)$

b) $NO_2(NN_{OMI}, \omega_0=0.95)$

**True $\boldsymbol{\omega}_0$ = 0.9**

c) $NO_2(NN_{True}, \omega_0=0.9)$

d) $NO_2(NN_{OMI}, \omega_0=0.95)$

True ALH range:

| | |
|---|---|
| —— | 0.3 [km] |
| —— | 0.5 [km] |
| —— | 1.4 [km] |
| —— | 3.5 [km] |

**Figure 7.** Relative $N^v_{NO_2}$ biases after application of the explicit aerosol correction as a function of true aerosol optical thickness (AOT) $\tau$(550 nm) and based on synthetic cases of Fig. 2. No bias is included on $\omega_0$: i.e. true and assumed values are identical: **(a)** True $\tau$ and $\omega_0$=0.95, **(b)** Retrieved OMI $\tau$ and $\omega_0$=0.95, **(c)** True $\tau$ and $\omega_0$=0.9, **(d)** Retrieved OMI $\tau$ and $\omega_0$=0.9.





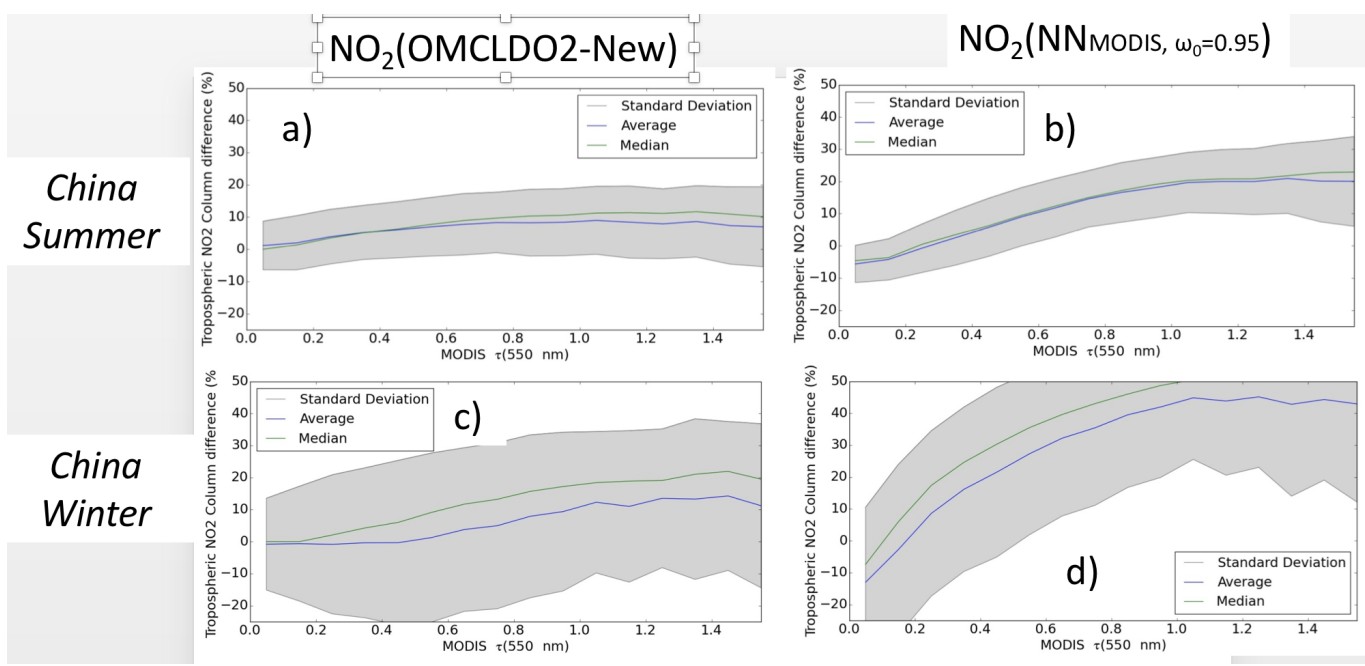

**Figure 8.** Statistic of relative $N_{NO_2}^v$ differences in (%) over China 2006-2007 in Summer (June-July-August) and Winter(December-January-February), after implicit or explicit aerosol correction compared to no aerosol correction (i.e. aerosol-free scene assumption): **(a)** Implicit aerosol correction based on OMCLDO2-New, **(b)** Explicit aerosol correction based on $NN_{MODIS,\omega_0=0.95}$.







**Figure 9.** Statistics of relative $N_{NO_2}^v$ changes in (%) in 2006-2007, due to differences between the different explicit aerosol corrections (see Tab. 1) and the implicit aerosol correction based on OMCLDO2-New: **(a)**, and **(b)**: China summertime (June-July-August), **(c)**, and **(d)**: China wintertime (December-January-February), **(e)**, and **(f)**: South America biomass burning season (August-September).



**Figure 10.** Maps of $c_p$ (converted into cloud height) from OMCLDO2-New and ALH from $NN_{MODIS, \omega_0=0.95}$ in [km] in 2006-2007: **(a)** $c_p$, China Summer, **(b)** ALH, China Summer, **(c)** $c_p$, China Winter, **(d)** ALH, China Winter, **(e)** $c_p$, South America, **(f)** ALH, South America.

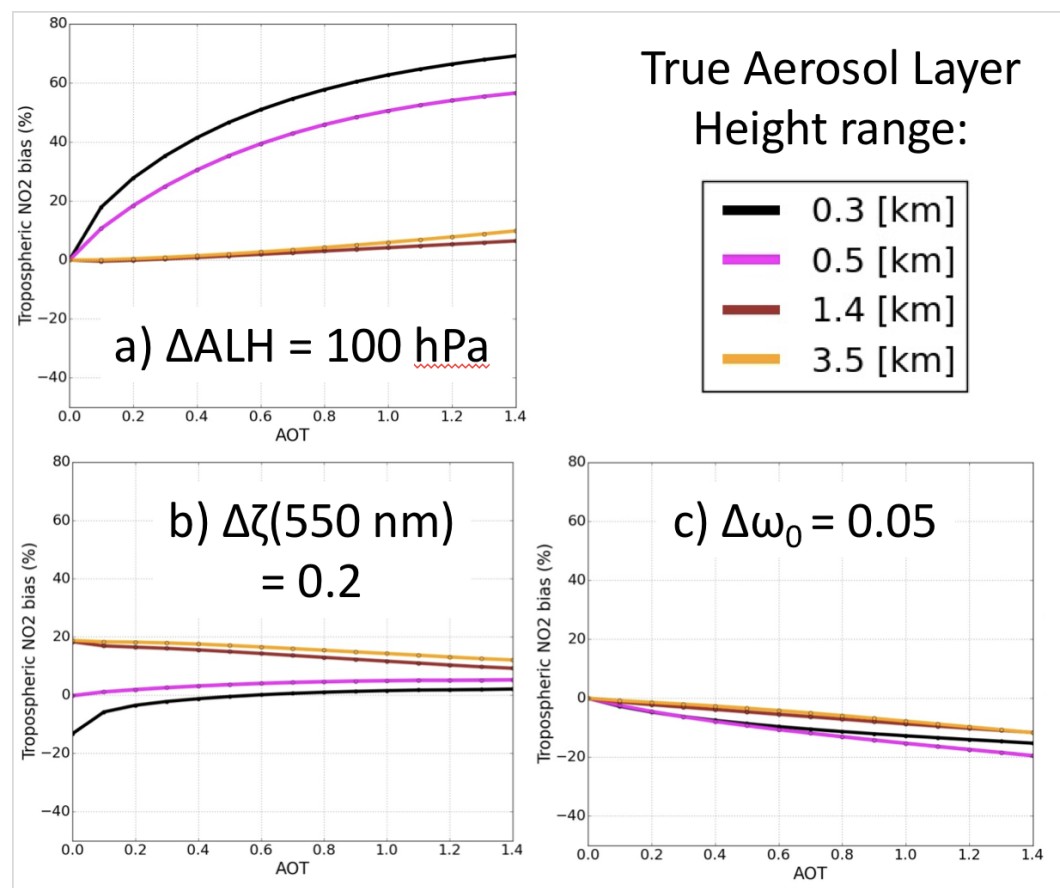

**Figure 11.** Relative $N_{NO_2}^v$ biases due to a direct impact of individual aerosol parameter biases on $A_{NO_2}^v$ (439 nm) calculation. Synthetic cases are similar to Fig. 2, except a typical mid-latitude winter NO₂ profile is considered instead: **(a)** ALH bias = 100 hPa, **(a)** $\tau$(550 nm) bias = 0.2, **(c)** $\omega_0$ bias = 0.05.





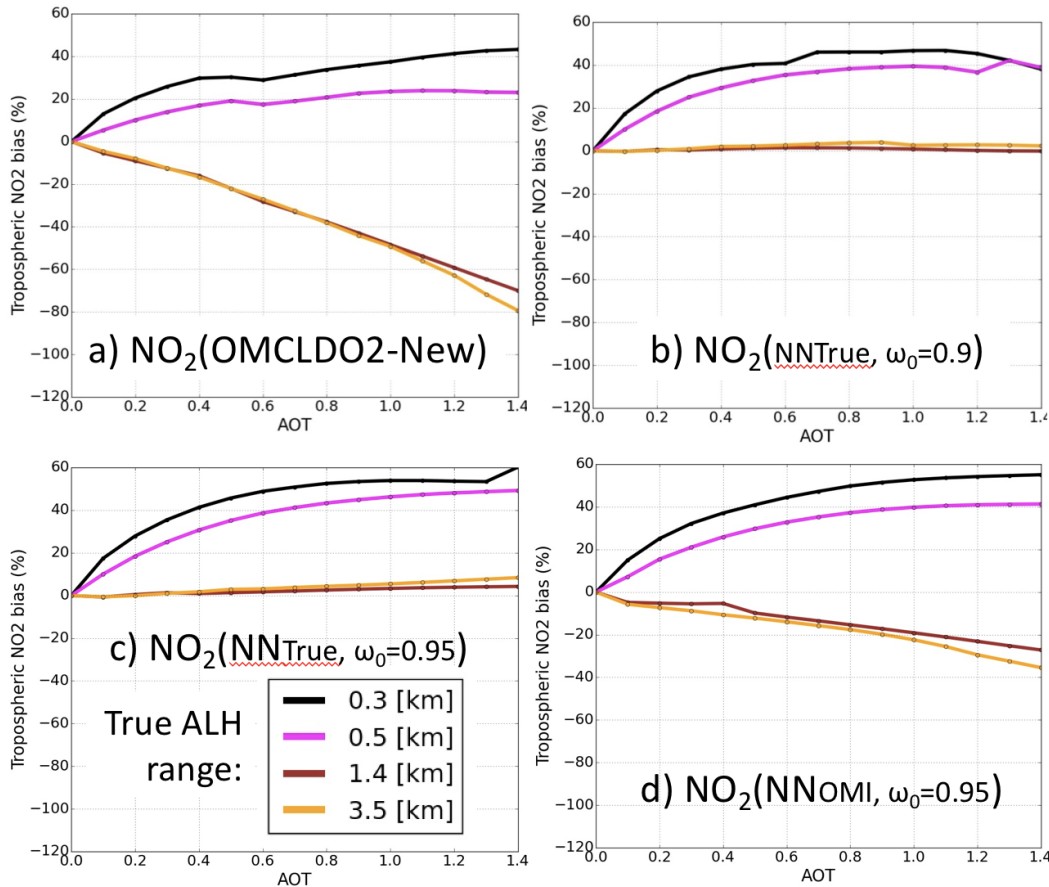

**Figure 12.** Relative $N_{NO_2}^v$ biases after application of the implicit or explicit aerosol correction (see Tab. 2) as a function of true aerosol optical thickness (AOT) $\tau(550\ nm)$. Synthetic cases are similar to Fig. 2 except the summer $NO_2$ profile was replaced by a typical winter one. Furthermore, only absorbing aerosols with true $\omega_0$=0.9 are considered. Finally, the impact of a bias in the assumed $\omega_0$ through the application of the explicit aerosol correction is illustrated: **(a)** Implicit aerosol correction based on OMCLDO2-New, **(b)** Explicit aerosol correction, true $\tau$ and assumed $\omega_0$=0.9, **(c)** Explicit aerosol correction, true $\tau$ and assumed $\omega_0$=0.95, **(d)** Explicit aerosol correction, retrieved OMI $\tau$ and assumed $\omega_0$=0.95.



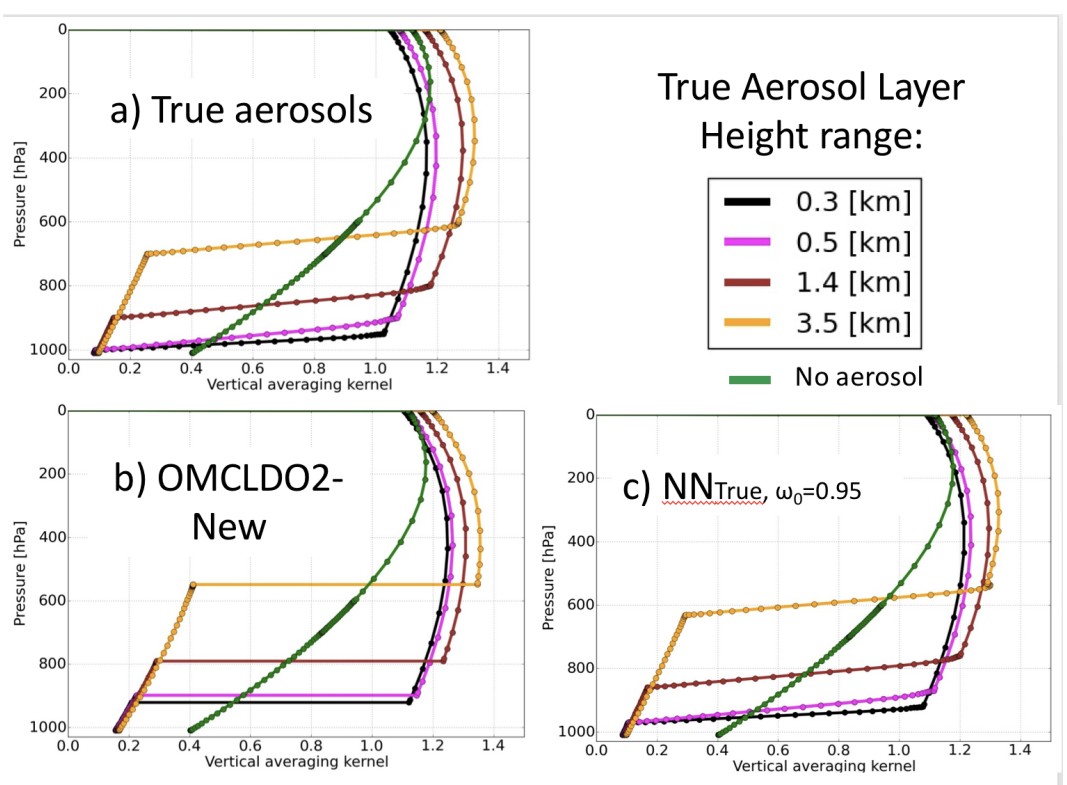

**Figure 13.** Vertical AK (cf. Sect. 3.2 and Sect. 5.4) based on aerosol $\tau(550\ nm) = 1.0$, $NO_2$ vertical profile of 1st of July, 2006 at 12:00 pm over China. Other conditions are similar to Fig. 2.





**Table 1.** Summary of the different OMI $N_{NO_2}^v$ mentionned in this paper, and the configuration of the associated aerosol correction: input OMI Lambertian Equivalent Reflectivity (LER) climatology, OMI cloud look-up table (LUT) for the effective cloud retrievals, Aerosol parameters (see for more details Sect. 2.2, 2.3 and 3)

| OMI $N_{NO_2}^v$ dataset | Aerosol correction | Configuration details |
|---|---|---|
| $NO_2$(DOMINO) | Implicit aerosol correction, OMCLDO2 | Coarse OMI cloud LUT |
| | | LER 3-year climatology |
| | | No temperature correction on $N_{O_2-O_2}^s$ |
| $NO_2$(OMCLDO2-Old) | Implicit aerosol correction, OMCLDO2 | Coarse OMI cloud LUT |
| | | LER 5-year climatology |
| | | Temperature correction on $N_{O_2-O_2}^s$ |
| $NO_2$(OMCLDO2-New) | Implicit aerosol correction, OMCLDO2 | Fine OMI cloud LUT |
| | | LER 5-year climatology |
| | | Temperature correction on $N_{O_2-O_2}^s$ |
| $NO_2(NN_{MODIS,\omega_0=0.9})$ | Explicit aerosol correction, OMI Aerosol NN | MODIS $\tau$(550 nm), OMI ALH |
| | | LER 5-year climatology |
| | | Assumed $\omega_0$=0.9 |
| | | Temperature correction on $N_{O_2-O_2}^s$ |
| $NO_2(NN_{OMI,\omega_0=0.9})$ | Explicit aerosol correction, OMI Aerosol NN | OMI $\tau$(550 nm), OMI ALH |
| | | LER 5-year climatology |
| | | Assumed $\omega_0$=0.9 |
| | | Temperature correction on $N_{O_2-O_2}^s$ |
| $NO_2(NN_{MODIS,\omega_0=0.95})$ | Explicit aerosol correction, OMI Aerosol NN | MODIS $\tau$(550 nm), OMI ALH |
| | | Assumed $\omega_0$=0.95 |
| | | LER 5-year climatology |
| | | Temperature correction on $N_{O_2-O_2}^s$ |
| $NO_2(NN_{OMI,\omega_0=0.95})$ | Explicit aerosol correction, OMI Aerosol NN | OMI $\tau$(550 nm), OMI ALH |
| | | Assumed $\omega_0$=0.95 |
| | | LER 5-year climatology |
| | | Temperature correction on $N_{O_2-O_2}^s$ |





**Table 2.** Summary of the changes in the diverse reprocessing OMI tropospheric $NO_2$ VCD $N_{NO_2}^v$ depending on the applied aerosol correction strategy (see Tab. 1) over all collocated MODIS aerosol scenes (MODIS $\tau(550 \text{ nm}) \geq 0.$). See more analyses in Sect. 4.1-4.2:

| Focus on: | Comparison reprocessed $NO_2$ | Region - Season | Changes in $N_{NO_2}^v$ in [%] Average $\pm$ Standard Deviation |
|---|---|---|---|
| Implicit correction | $NO_2$(OMCLDO2-Old) $-$ $NO_2$(DOMINO) | China - summer | $-1.0 \pm 9.0$ |
| | | China - winter | $-15.6 \pm 29.8$ |
| | | South America - biomass burning | $-6.2 \pm 16.0$ |
| | $NO_2$(OMCLDO2-New) $-$ $NO_2$(OMCLDO2-Old) | China - summer | $1.3 \pm 6.9$ |
| | | China - winter | $7.9 \pm 19.3$ |
| | | South America - biomass burning | $7.9 \pm 14.4$ |
| | $NO_2$(OMCLDO2-New) $-$ $NO_2$(DOMINO) | China - summer | $0.4 \pm 10.6$ |
| | | China - winter | $-4.0 \pm 26.8$ |
| | | South America - biomass burning | $3.1 \pm 17.3$ |
| . Explicit correction | $NO_2(NN_{MODIS,\omega_0=0.95}) - NO_2$(OMCLDO2-New) | China - summer | $-2.9 \pm 12.5$ |
| | | China - winter | $6.8 \pm 26.1$ |
| | | South America - biomass burning | $-8.1 \pm 16.8$ |
| | $NO_2(NN_{MODIS,\omega_0=0.9}) - NO_2(NN_{MODIS,\omega_0=0.95})$ | China - summer | $-0.2 \pm 7.8$ |
| | | China - winter | $-8.2 \pm 22.3$ |
| | | South America - biomass burning | $1.3 \pm 8.7$ |
| | $NO_2(NN_{OMI,\omega_0=0.95}) - NO_2$(OMCLDO2-New) | China - summer | $6.5 \pm 11.9$ |
| | | China - winter | $11.2 \pm 18.4$ |
| | | South America - biomass burning | $-3.0 \pm 14.0$ |
| | $NO_2(NN_{OMI,\omega_0=0.9}) - NO_2(NN_{OMI,\omega_0=0.95})$ | China - summer | $8.5 \pm 13.7$ |
| | | China - winter | $-1.9 \pm 24.3$ |
| | | South America - biomass burning | $1.0 \pm 14.6$ |