# Peer review of "Minimizing aerosol effects on the OMI tropospheric NO₂ retrieval – An improved use of the 477 nm O₂-O₂ band and an estimation of the aerosol correction uncertainty"

_Atmospheric Measurement Techniques, 2018_

## Referee Comment (RC1) · Anonymous Referee #1 · 5 Sep 2018

Review of "Minimizing aerosol effects on the OMI tropospheric NO2 retrieval – An improved use of the 477 nm O2 -O2 band and an estimation of the aerosol correction uncertainty" by Chimot et al.

This paper describes results of applying the aerosol correction to the OMI operational NO2 algorithm, DOMINO-v2. The aerosol correction is applied in two different forms: the implicit aerosol correction using the improved OMI operational O2-O2 cloud algorithm, OMCLDO2 v2, and the explicit correction using aerosol parameters derived with an aerosol neural network (NN). The authors conclude that both approaches to the aerosol correction reduce the biases identified in DOMINO-v2 over polluted cloud-free areas. The paper contains significant original material that can be of interest for the developers of cloud and trace gas algorithms for satellite sensors. The paper subject is appropriate to AMT. Earlier work is adequately recognized and credited. The abstract provides a sufficiently complete summary of the paper. I recommend the paper for publication after the authors address the following comments.

**General comments**

1. The aerosol model used in simulations assumes a single value of the asymmetry parameter for the Henyey-Greenstein (HG) phase function and a single value of the Angstrom exponent. The model seems to be oversimplified. There is no justification for the selection of these parameters. To support the choice of the HG function, the authors reference the paper by Dubovik et al. (2002). However this paper does not discuss the HG phase function. Additionally, the assumed aerosol model is not consistent with an aerosol model used in the MODIS aerosol retrieval algorithms while the authors propose to combine the MODIS AOT retrievals with their aerosol model in the explicit aerosol correction (see P.8, L. 14).

2. The authors consider absorbing aerosols only with relatively low values of the single scattering albedo (SSA) of 0.9 and 0.95. Maps of the seasonal mean SSA at 500 nm retrieved from OMI from 2005 to 2016 (Fig. 5, Kang et al., Remote Sensing, 9, 1050, 2017) show that SSA values over China are greater than 0.96-0.98  in summertime and mostly greater than 0.92-0.94 in wintertime. The authors should redo their calculations shown in Fig. 4 with more realistic values of SSA for summertime over China.

3. In Sect. 5.5, the authors discuss an important issue of the TOA radiance closure in case of using the MODIS AOT retrievals with OMI-derived climatological surface LER. The authors correctly state that OMI-derived LER is inconsistent with the surface BRDF used in the MODIS aerosol algorithm. Moreover, there are significant differences in absolute values of the OMI-derived LER and MODIS atmospherically corrected surface reflectance/albedo. Those differences are due to that the climatological LERs include a contribution from inevitable aerosol

contamination and possible cloud contamination for OMI pixels which are much larger than MODIS pixels. Because of this essential inconsistency I would suggest to consider dropping the use of the MODIS AOT in the explicit aerosol correction.

4. Results of the use of the explicit aerosol correction are shown for averages over 3 months (see Fig. 4-6). However, it remains unclear how to use the explicit aerosol correction operationally, on an orbit by orbit basis. It would be quite beneficial for a reader to provide practical recommendations for operational processing with the explicit aerosol correction.

5. There are many typos in the manuscript. Some of them are listed in Technical notes below.

**Specific comments**

P.3, L.22. Please consider adding references to the papers published by other research groups, for instance:

Joiner et al., Retrieval of cloud pressure and chlorophyll content using Raman scattering in GOME ultraviolet spectra. J. Geophys. Res., Vol.109, D01109, doi:10.1029/2003JD003698, 2004.

Vasilkov et al., A cloud algorithm based on the O2-O2 477 nm absorption band featuring an advanced spectral fitting method and the use of surface geometry-dependent Lambertian-equivalent reflectivity, Atmos. Meas. Tech., 11, 4093-4107, doi:10.5194/amt-10-4093-2018, 2018.

P.5, L.13. There are contributions of ozone absorption and Raman scattering to the top-of-the-atmosphere radiance at 475 nm. How the ozone absorption and Raman scattering are accounted for in the definition of the continuum reflectance at 475 nm?

P.5, L.23-24. The surface reflectance is considered "as a second order effect" on the O2-O2 slant column density. This must be justified.

P.7, L.29. Please clarify "the nature of the O2-O2 spectral band". What do you mean?

P.11, Fig. 2. The NO2 bias dependence on AOT is quite non-monotonic. It looks too "bumpy" to be real. Please explain so strange behavior of the curves in Fig. 2.

P.11, Last paragraph. Data in Fig. 3 are shown for AOT(550) up to 1.5 which corresponds to AOT(475) of about 1.9. So high values of AOT may lead to effective cloud fractions exceeding the threshold of 0.1 that was stated on Page 8, Line 29. Please provide values of effective cloud fraction for those data.

P.11, L.14-17. Just two sentences describe Fig. 4-6. They deserve more lengthy discussion.

P.13, Fig. 9 discussion. Please explain why there is so big difference in the behavior the light blue (SSA=0.95) and green (SSA=0.9) curves in Fig. 9a and Fig. 9b. The curves are close to each other in Fig. 9a and they are dramatically different in Fig. 9b.

P.14, L.16. There is no discussion about ALH in Sect. 3.2.

P.15, L.30. The authors state that the single scattering albedo (SSA), the asymmetry parameter, and the Angstrom exponent are "of second importance". This statement should be justified. Please explain why calculations are carried out for two values of SSA and for a single value of the asymmetry parameter and the Angstrom exponent? Are they "of third importance"?

P.16, L.4-5. It is quite desirable to show NO2 profiles used in calculations.

P.16, L.28. Please clarify the meaning of "an insufficient coverage of the observation scene".

P.17, Sect.5.4. Please provide a value of the effective cloud fraction in Fig. 13b.

P.34. No captions for Fig. 4c and 4d.

P.35. Fig. 9d, 9e, and 9f are missing while they are mentioned in the figure caption.

**Technical notes**

P.1, L.4. "Minimizing … are", should be "is"?

P. 2, L. 8. "health population". What do you mean?

P.2, L.10. Should be "Ozone".

P.2, L. 18. "mapping ... have", should be "has"?

P.3, L. 5-8.It is hard to follow. Please reword or split the sentence.

P.3, L.7. Should be "Orthogonal"

P.4, L.21. A typo. Should be "477"

P.5, L.9. Remove "x"

P.6, L.4. and elsewhere, e.g. P.7, L.28; P.11, L.14; P.14, L.12. Should be subscripts.

P.10, L9. "One the main". "of" is missing.

P.10, L.17. A typo in "information"

P.11, L.33. "attenuate the biases". Do you mean "reduce"?

P.13, L.33-34. It may be hard to understand this sentence. Please clarify.

P.13, L.34. "This may of course be …". What is "this"?

P.16, L.29. Remove period after "scene"

P.16,L.31. Should be "potential"?

P.17, L.4. NO2 vertical column "degradations are more important". Please reword.

P.17, L.18. Should be 550 nm?

P.17, L.25-26. You may want to reword "the transmission of the clear fraction of the pixel through the IPA assumption". Otherwise, it is not clear.

P.30. Fig. 4 and elsewhere in other figures. The Greek symbol "tau" in Fig. 4a is misspelled.

P.37. Figure caption. Second (a) should be (b).

---

## Referee Comment (RC2) · Anonymous Referee #2 · 24 Sep 2018

The comment was uploaded in the form of a supplement:
https://www.atmos-meas-tech-discuss.net/amt-2018-247/amt-2018-247-RC2-supplement.pdf

---

## Author Comment (AC2) · 12 Dec 2018

The comment was uploaded in the form of a supplement:
https://www.atmos-meas-tech-discuss.net/amt-2018-247/amt-2018-247-AC2-supplement.pdf

---

## Author Comment (AC1)

**Interactive discussion on AMTD-2017-286 "Minimizing aerosol effects on the OMI tropospheric NO2 retrieval – An improved use of the 477 nm O2-O2 band and an estimation of the aerosol correction uncertainty"**

Julien Chimot et al.

**Julien.Chimot@eumetsat.int**

We would like to thank very much Referee #1 for his / her valuable comments. They give us the opportunity to solidify our messages and manuscript. Below we address them one by one (Referee #1 comments in blue, author and co-authors in black).

**General comments**

This paper describes results of applying the aerosol correction to the OMI operational NO2 algorithm, DOMINO-v2. The aerosol correction is applied in two different forms: the implicit aerosol correction using the improved OMI operational  $O_2$ - $O_2$  cloud algorithm, OMCLDO2 v2, and the explicit correction using aerosol parameters derived with an aerosol neural network (NN). The authors conclude that both approaches to the aerosol correction reduce the biases identified in DOMINO-v2 over polluted cloud-free areas. The paper contains significant original material that can be of interest for the developers of cloud and trace gas algorithms for satellite sensors. The paper subject is appropriate to AMT. Earlier work is adequately recognized and credited. The abstract provides a sufficiently complete summary of the paper. I recommend the paper for publication after the authors address the following comments.

**We tried to take as much as possible your comments below. Please find below our answers and clarifications where requested.**

**General comments**

1. The aerosol model used in simulations assumes a single value of the asymmetry parameter for the Henyey-Greenstein (HG) phase function and a single value of the Angstrom exponent. The model seems to be oversimplified. There is no justification for the selection of these parameters. To support the choice of the HG function, the authors reference the paper by Dubovik *et al.* (2002). However, this paper does not discuss the HG phase function. Additionally, the assumed aerosol model is not consistent with an aerosol model used in the MODIS aerosol retrieval algorithms while the authors propose to combine the MODIS AOT retrievals with their aerosol model in the explicit aerosol correction (see P.8, L. 14).

The choice of the HG is thoroughly discussed in Chimot *et al.* (2017, 2018), and briefly reminded in the present manuscript in Sect. 2.3. The main motivation of the exploratory development of an aerosol layer height (ALH) retrieval algorithm, using the OMI 477 nm  $O_2$ - $O_2$  absorption band, has been the aerosol correction in the visible spectral range in view of tropospheric NO2 retrieval. In Chimot *et al.* (2016), we quantitatively demonstrated that, for such a purpose, AOD and ALH are the key parameters needed. Other aerosol parameters, that are more related to their optical properties, shape, and size are of a second importance. This is supported by a number of studied such as (Boersma *et al.*, 2004; Leitao *et al.*, 2010; Castellanos *et al.*, 2015; Chimot *et al.*, 2016). The main reasons are because to correct of aerosol effects, we overall need the length of the average light path in presence of scattering and absorbing particles. This is primarily driven by AOD and ALH (in addition to the shape of the NO2 vertical profile), much less by the detailed properties of particles. You can see more of our discussion in our previous publications (Chimot *et al.*, 2016, 2017). Therefore, we concluded that AOD and ALH are the first parameters primarily needed for an aerosol correction of the tropospheric NO2 AMF, and other details describing the shape of the scattering phase function are of second importance, even if not negligible.

The HG phase function is commonly used in several reference studies focusing on aerosol correction for trace gases: *e.g.* Wagner *et al.*, 2077; Vlemmix *et al.*, 2010; Castellanos *et al.*, 2015. Supporting by the recommended value of g = 0.7 by Dubovik *et al.* (2002), the HG model does not seem then oversimplified in this context as it is known to be smooth and reasonable reproduces the Mie scattering phase functions. However, we do acknowledge that using Mie Scattering (for example) would be more

accurate for spherical particles with more details (in particular with respect to the "tail" of the backward scattering direction). But the gain in accuracy would be lower compared to crucial issue of using accurate and representative AOD and ALH values for every single OMI measurement pixel.

Furthermore, retrieving ALH from passive hyperspectral sensor ideally requires aerosol type information (and vertical profile shape) for every single measured scene / pixel. However, such an accurate information is not available at such a scale. Therefore, we do need to make assumptions. One of the most common approaches is to assume one aerosol type model which should be representative, in average, of the most common aerosol types and mixtures encountered by the space-borne measurements. The general approach, in an operational context, is the use of the HG function for ALH retrieval from the Sentinel-4, Sentinel-5 and Sentinel-5 Precursor sensors (Sanders et al., 2015, Nanda et al., 2017, 2018), although based on the  $O_2$ -A band.

We have evaluated the performance of our retrieval, based on this assumption, over China, South America, and Russia areas with scenes including urban, industrial, and biomass burning pollution events and for different seasons (Chimot *et al.*, 2017, 2018). These scenes are mostly dominated by fine spherical particles, weakly absorbing (*e.g.* sulfate, and nitrate) or strongly absorbing (*e.g.* smoke). Dust particles may sometimes be mixed. We overall showed the good performance of the retrievals.

The impact of the assumed phase function is shown to be critical mostly when pure coarse dust aerosol signal dominate in the measured spectrum (*i.e.* not mixed with other types of particles). Such aerosols are known to be mostly spheroid and irregularly shaped. Their phase function shall be then better represented by T-Matrix models.

However, scenes over China, America, Africa, and Europe with high NO2 pollution are mostly dominated by spherical fine weakly and strongly absorbing. Therefore, the HG function is assumed to be, for these scenes, correct at a first order.

As a natural follow-up of our previous works, one of the motivations of this paper is to present an exploratory study where, for the first time, we show the possibility to apply an explicit aerosol correction based on aerosol parameters derived from the 477 nm  $O_2$ - $O_2$  absorption band from the same instrument. We made clearer, at several places in our revised manuscript, that this study is an exploratory work evaluating the feasibility of a first explicit aerosol correction from  $O_2$ - $O_2$  measurements acquired by a same sensor simultaneously with the NO2 absorption spectral band.

For the discussion of combining with MODIS dataset, please see our answers further in the document.

2. The authors consider absorbing aerosols only with relatively low values of the single scattering albedo (SSA) of 0.9 and 0.95. Maps of the seasonal mean SSA at 500 nm retrieved from OMI from 2005 to 2016 (Fig. 5, Kang et al., Remote Sensing, 9, 1050, 2017) show that SSA values over China are greater than 0.96-0.98 in summertime and mostly greater than 0.92- 0.94 in wintertime. The authors should redo their calculations shown in Fig. 4 with more realistic values of SSA for summertime over China.

The two typical single scattering albedo (SSA) values have been assumed in Chimot *et al.* (2017, 2018). They are based on the fact that aerosol pollution in dense urban and industrialized areas in China, in summertime, are dominated by a mix of weakly absorbing particles (*e.g.* sulfate, nitrate) and dust depending on the event transport from the surrounding deserts. However, in summertime in the same area, dust transport is generally minimal (at least less than in Spring), and weakly absorbing particles are usually present in high abundance due to the urban and industrial activities relying on energy production from (coal) power plants. Furthermore, wildfires in regions like South America release heavy load of very absorbing aerosols (*e.g.* smoke with black soot).

Reliable SSA values, in our knowledge, are better derived from ground-based sensors, such as the AERONET network. In spite of multiple efforts, it remains one of the most difficult aerosol parameters to estimate from satellite sensors, in particular from passive instruments. A lot of studies show SSA values that are retrieved from OMI as mentioned by Referee #1. One of the best achievements is from the near-UV two-channel algorithm (OMAERUV), described in a lot of studies (Torres *et al.*, 2002, 2007, 2013). This algorithm relies on the absorption property of aerosols, and its competition with the large Rayleigh scattering signal, in the UV spectral range. However, in spite of its robustness, this algorithm critically depends on the aerosol layer height (ALH), cloud residuals, and surface albedo assumptions. All of these elements are not easy to handle for each single OMI pixel, especially for ALH which is not available on a daily basis for every single OMI pixel. Jethva *et al.* (2014) intercompared the OMI SSA

values with AERONET at multiple locations, and showed that absolute differences are in the range of  $\pm$  0.03 ( $\pm$  0.05) for 49% (AOD < 0.7) and 53% (AOD > 0.7). But, inconsistencies are observed for biomass burning retrievals over Southeast Asia and Australia, and over multiple urban/industrial sites, including China, at moderate and low aerosol loading, AOD(440 nm) < 0.7. In particular, it is found that retrieved SSA values are overestimated. And this is likely the case in the Fig.5 shown in Kang et al. Lin *et al.* (2014, 2015) showed typical aerosol SSA values over China from the GEOS-Chem model: on average, values are in the range of 0.95-0.96 over China during summer time, and more around 0.9 in winter time.

Lin, J.-T., Martin, R. V., Boersma, K. F., Sneep, M., Stammes, P., Spurr, R., Wang, P., Van Roozendael, M., Clémer, K., and Irie, H.: Retrieving tropospheric nitrogen dioxide from the Ozone Monitoring Instrument: effects of aerosols, surface reflectance anisotropy, and vertical profile of nitrogen dioxide, Atmospheric Chemistry and Physics, 14, 1441–1461, doi:10.5194/acp-14-1441-2014, 2014.

Lin, J.-T., Liu, M.-Y., Xin, J.-Y., Boersma, K. F., Spurr, R., Martin, R., and Zhang, Q.: Influence of aerosols and surface reflectance on satellite NO2 retrieval: seasonal and spatial characteristics and implications for  $NO_x$  emission constraints, Atmospheric Chemistry and Physics, 15, 11 217–11 241, doi:10.5194/acp-15-11217-2015, 2015.

Overall, we would like to stress that our exploratory study of applying an explicit aerosol correction based on simultaneous measurements from the 477 nm  $O_2$ - $O_2$  absorption band is by nature limited to the accuracy of the aerosol model, and most especially the assumed aerosol type. One could do multiple calculations with an infinite number of SSA values as well as the angstrom parameter or the asymmetry parameter g. However, that would primarily impact the derivation of ALH and consequently, the tropospheric  $NO_2$  AMF. We think that here we demonstrate the possibility of such an explicit aerosol correction. The impact of changing the assumed aerosol model in the forward model, through the ALH (and AOD) retrieval and, consequently the tropospheric NO2 AMF computation is discussed in details in our manuscript (Sect. 5 and Fig. 12).

However, to confirm the real performances of the suggested explicit aerosol correction strategy here and to move from this exploratory phase to an operational processing of OMI data (and even future sensors), new NN algorithms should be designed and trained with a larger dataset that includes accurate aerosol parameters (size and  $\omega$ 0) combined with different detailed models of the phase function. These models shall include reference Mie scattering (for spherical particles) and T-Matrix (for dust) to account accurately for all the details in the scattering phase function, size, and amount of multiple scattering effects. Each of these algorithms should be evaluated on a high number of specific observations to conclude on the exact aerosol model type to be assumed for the OMI visible spectral measurements. This follows the natural recommendations of our ALH retrieval analyses in Chimot *et al.* (2018). However, the accuracy of the derived parameters will then critically depend on the feasibility to know a priori the aerosol type (or the mixing). Such an exercise is out the scope of the present paper, and should be subject of a new study with dedicated analyses.

3. In Sect. 5.5, the authors discuss an important issue of the TOA radiance closure in case of using the MODIS AOT retrievals with OMI-derived climatological surface LER. The authors correctly state that OMI-derived LER is inconsistent with the surface BRDF used in the MODIS aerosol algorithm. Moreover, there are significant differences in absolute values of the OMI- derived LER and MODIS atmospherically corrected surface reflectance/albedo. Those differences are due to that the climatological LERs include a contribution from inevitable aerosol contamination and possible cloud contamination for OMI pixels which are much larger than MODIS pixels. Because of this essential inconsistency I would suggest to consider dropping the use of the MODIS AOT in the explicit aerosol correction.

We understand the motivation of the Referee 1 here. Dropping the results based on MODIS AOT is naturally tempting given our thoughts about the inconsistencies and the issue of the TOA radiance radiance closure budget. However, we think that such an issue is likely only obvious now after reading our results and analyses. We don't think this is naturally obvious for everyone prior to this. One reason is that the accuracy of the ALH retrieval is strongly dependent on the requested prior AOT. We demonstrated with Chimot *et al.* (2017, 2018) that the most accurate ALH retrievals were obtained with MODIS AOD, not our own derived OMI AOT. This naturally suggest first that the combination of OMI ALH – MODIS AOT shall give the most accurate tropospheric NO2 AMF. Dropping in our paper the use of MODIS AOT due to the TOA radiance closure budget may sound, at a first view, for any reader, confusing and probably counter-intuitive. Also, we have noticed some studies consider inga mix of different satellite observations and model (for *e.g.* CALIOP aerosol profiles, GEOS-Chem model, and MODIS AOD) while the issue of radiance closure budget is not clear.

We would then prefer to keep it, with our referred analyses. We wish to emphasize that the TOA radiance closure budget and the AMF accuracy are 2 distinctive aspects and require different (opposite?) strategies. Eventually, the best strategy is, in our opinion, not straightforward. And it requires more discussions in our community and investigations to understand how to proceed further.

Our past results do not necessarily reflect that inconsistency with the MODIS aerosol models critically bias our retrievals. However, it is true this may lead to some more insidious impacts, and we mention here such that closure of the radiance TOIA budget. Considering the MODIS aerosol models would not be a straightforward operation, as 1) one should think about the reliability of the detected most probably aerosol type per MODIS AOD retrieval, 2) one would need all the details of the models, including ancillary parameters such as surface reflectance, and how one could translate them correctly to the OMI spectral observations configurations (taking into account the differences in geometry and spectral response functions).

4. Results of the use of the explicit aerosol correction are shown for averages over 3 months (see Fig. 4-6). However, it remains unclear how to use the explicit aerosol correction operationally, on an orbit by orbit basis. It would be quite beneficial for a reader to provide practical recommendations for operational processing with the explicit aerosol correction.

The results are indeed shown *via* statistics of the overall tropospheric NO2 retrievals, after implicit or explicit aerosol corrections, and their differences. But the corrections were applied per individual OMI pixel / NO2 retrieval, for each single measurement identified as cloud-free according to the criteria described in Sect. 3.1. The explicit aerosol correction was thus applied as the following, for each single measurement: 1) ALH (and AOD) were retrieved from the OMI 477 nm O2-O2 spectral band, 2) these parameters were applied to compute the tropospheric NO2 AMF with the DISAMAR radiate transfer model, 3) the AMF was applied to the DOMINO NO2 slant column density (SCD) to obtain the tropospheric NO2 vertical column density (VCD).

As such, the applied explicit aerosol correction was applied in an almost operational working mode. Furthermore, it is worth reminding that the developed neural network (NN) technique, once the training is finished and validated, allows a very fast computing time, a big advantage in a context of big data challenge.

For an operational processing on an orbit by orbit basis, the following elements are recommended prior to the explicit correction: 1) cloudy pixels need to be detected and screened, such that the explicit aerosol correction is not allowed, 2) other prior input parameters shall be accessible (*e.g.* surface albedo), 3) ALH (and AOD) retrievals need to be done prior to the NO2 retrieval. A non-negligible issue remains however open with cloudy (or mixed aerosol-cloud) cases. The explicit aerosol correction being not developed for such a purpose, the question remains open whether they should be processed with effective clouds, or just filtered out. This may introduce some discontinuities in the processed datasets. Such an issue shall be strongly investigated, but is clearly beyond the scope of this paper.

**5. There are many typos in the manuscript. Some of them are listed in Technical notes below.**

We sincerely apologize for the typos left in our submitted discussion manuscript. We took into all your suggestions below. We then performed a new and thorough review of our writing. We hope the writing has been well improved.

**Specific comments**

P.3, L.22. Please consider adding references to the papers published by other research groups, for instance:

Joiner *et al.*, Retrieval of cloud pressure and chlorophyll content using Raman scattering in GOME ultraviolet spectra. J. Geophys. Res., Vol.109, D01109, doi:10.1029/2003JD003698, 2004.

Vasilkov et al., A cloud algorithm based on the O2-O2 477 nm absorption band featuring an advanced spectral fitting method and the use of surface geometry-dependent Lambertian- equivalent reflectivity, Atmos. Meas. Tech., 11, 4093-4107, doi:10.5194/amt-10-4093-2018, 2018.

Thanks for your suggestions, we added them where appropriate. Please verify in our revised manuscript.

P.5, L.13. There are contributions of ozone absorption and Raman scattering to the top-of-theatmosphere radiance at 475 nm. How the ozone absorption and Raman scattering are accounted for in the definition of the continuum reflectance at 475 nm?

The continuum reflectance at 475 nm is defined as the reflectance that would be measured without any gas absorption.

Ozone absorption and Raman scattering were considered in the generation of the simulations for the training dataset of the neural networks in our previous studies. Similarly to the OMI effective cloud algorithm, a DOAS fit is first applied in the 460-490 nm OMI band to derive the continuum reflectance and the  $O_2$ - $O_2$  slant column density by relying on the Beer-Lambert law. To "remove" the ozone absorption, the ozone slant column density was also fitted together with the  $O_2$ - $O_2$ . However, it is not used at the end for the conversion, *via* the neural networks, into aerosol parameters (ALH and AOD).

**P.5, L.23-24. The surface reflectance is considered "as a second order effect" on the $O_2$ - $O_2$ slant column density. This must be justified.**

We clearly demonstrated in Chimot *et al.* (2017), in particular with Figure 1 and reference radiative transfer simulations, that the  $O_2$ - $O_2$  slant column density is primarily driven by AOD and ALH. Aerosol properties such as SSA, or surface reflectance, also impact the  $O_2$ - $O_2$  absorption but as a second order effect. Note that we also demonstrated that the uncertainties of our ALH retrievals were primarily driven by aerosol parameters such as AOD and SSA. Surface albedo uncertainty had a lower impact, up to 200 m.

**P.7, L.29. Please clarify "the nature of the O2-O2 spectral band". What do you mean?**

We refer here to the low signal of the  $O_2-O_2$  absorption band at 477 nm, about 2-3%. This explains that low aerosol load (*i.e.* low AOD) have little impacts on the  $O_2-O_2$  absorption shielding. This is why our ALH retrievals are only accurate for AOD(550 nm) > 0.5 (or even 0.6).

**P.11, Fig. 2. The $NO_2$ bias dependence on AOT is quite non-monotonic. It looks too "bumpy" to be real. Please explain so strange behavior of the curves in Fig. 2.**

In Fig.2, the aerosol correction is the implicit one, and derived from the OMI cloud LUTs. The interpolation, that derives the effective cloud parameters used for the tropospheric NO2 AMF computation, strongly depends on the sampling nodes in the given LUTs. In Figs. 2a and c, we used the old LUT. We demonstrated in Chimot *et al.* (2016) that the sampling of these LUTs was too coarse leading to numerical defaults and too much discontinuity in the NO2 bias dependence (a bit "bumpy). In Figs. 2b and d, the new LUTs were used with a much higher sampling / resolution, Not only, this solves the observed numerical defaults, corrects the previously observed underestimated tropospheric NO2 VCD, but also gives smoother NO2 bias dependence on AOT. However, we acknowledge some (more minor) discontinuities are still observed. It is interesting that, with the explicit aerosol correction in Fig 7, these discontinuities are almost completely removed. This may be linked to the apparently more continuous interpolation approach with the NNs leading to smoother NO2 bias dependence.

P.11, Last paragraph. Data in Fig. 3 are shown for AOT(550) up to 1.5 which corresponds to AOT(475) of about 1.9. So high values of AOT may lead to effective cloud fractions exceeding the threshold of 0.1 that was stated on Page 8, Line 29. Please provide values of effective cloud fraction for those data. P.11, L.14-17. Just two sentences describe Fig. 4-6. They deserve more lengthy discussion.

We confirm that collocated OMI-MODIS data are selected with a threshold of 0.1 on effective cloud fraction. There is then no higher values in the OMI effective cloud fraction for the selected observations. In (Chimot *et al.*, 2016), we demonstrated that the OMI effective cloud fraction represents the enhanced brightness of the OMI scene. This enhanced brightness is primarily due to the presence of scattering particles, and is thus strongly correlated with AOT. However, the scene brightness is not only

driven by AOT, but also by the surface albedo and other aerosol properties such as SSA. We showed in (Chimot *et al.*, 2016) that, for a same AOT value, the OMI effective cloud fraction is lower over a bright surface and higher over a dark surface. This is because aerosols, in spite of their more or less scattering nature, shield part of the surface reflectance. It is also worth emphasizing that, for a same AOT value, the OMI effective cloud fraction is higher for higher SSA values (*i.e.* no or weakly absorbing particles). Note also that observation geometry may also play a role.

Therefore, even if AOT values may be high in the visible, effective cloud fraction values can remain close to (or even below) 0.1 depending on surface conditions and aerosol properties. This threshold is a good way to filter out as much as possible potential cloud residuals, but it is possible that scenes with very scattering particles and thick optical thickness were excluded as well. Finding a perfect threshold value remains however very challenging.

**P.13, Fig. 9 discussion. Please explain why there is so big difference in the behavior the light blue (SSA=0.95) and green (SSA=0.9) curves in Fig. 9a and Fig. 9b. The curves are close to each other in Fig. 9a and they are dramatically different in Fig. 9b.**

There can be several reasons for that. The tropospheric  $NO_2$  AMF is dependent on the assumed scattering / absorbing model representing the simulated particles, but on the  $NO_2$  vertical profile shape. Depending on the  $NO_2$  vertical distribution along the vertical atmospheric layers, and the scattering *vs.* absorbing property of aerosols, part of the  $NO_2$  may be sampled by the photons along the light path. This sampling influences then the computed  $NO_2$  AMF. In summertime, the tropospheric  $NO_2$  bulk at the surface is much less important than in Winter.

**P.14, L.16. There is no discussion about ALH in Sect. 3.2.**

Correct, we initially discussed this in the introduction and did not repeat it. We added the following lines at the end of Section 3.2.:

"Among all these variables, many studies emphasized that ALH and  $\tau$  are the most critical aerosol parameters affecting primarily AvNO2 over cloud-free scenes dominated by aerosol particles (Leitão *et al.*, 2010; Castellanos *et al.*, 2015; Chimot *et al.*, 2016). It was clearly demonstrated that other parameters describing aerosol properties, such as size, are generally of second order of magnitude for such a purpose.".

P.15, L.30. The authors state that the single scattering albedo (SSA), the asymmetry parameter, and the Angstrom exponent are "of second importance". This statement should be justified. Please explain why calculations are carried out for two values of SSA and for a single value of the asymmetry parameter and the Angstrom exponent? Are they "of third importance"?

Please see our answers earlier in the document, with respect to your main comments.

**P.16, L.4-5. It is quite desirable to show NO2 profiles used in calculations.**

The NO2 profiles used in our calculations are those present in the DOMINO product for each NO2 slant column density. They vary per OMI pixel observation accordingly to the DOMINO product.

**P.16, L.28. Please clarify the meaning of "an insufficient coverage of the observation scene".**

We refer here to the effective cloud fraction parameter that determines the fraction of the OMI scene by the Lambertian cloud layer. In case of absorbing aerosols, the effective cloud fraction is smaller than. In presence of scattering particles due to the reduced scene brightness. Consequently, the cloud fraction or cloud coverage is reduced. We clarified this in Sect. 5.3.

**P.17, Sect.5.4. Please provide a value of the effective cloud fraction in Fig. 13b.**

In that case, effective cloud fraction values are for all scenes are very close to 0.1 (about 0.097) for all aerosol layer height (effective cloud fraction is almost not affected by ALH for a same AOD).

**P.34. No captions for Fig. 4c and 4d.**

Figs. 4c and 4d are actually already indicated in our caption.

P.35. Fig. 9d, 9e, and 9f are missing while they are mentioned in the figure caption.

Sorry, this is a typo in the caption. Only figures 9a, b, and c are indeed depicted here.

Technical notes P.1, L.4. "Minimizing ... are", should be "is"? Yes, corrected.

**P. 2, L. 8. "health population". What do you mean?**

Here, we refer to the fact that air quality observations, like from OMI or S5P, can verify the effectiveness of some implemented actions that are crucial in view of reducing air pollution (for environment and population protections). We clarified this.

**P.2, L.10. Should be "Ozone".**

Corrected.

P.2, L. 18. "mapping ... have", should be "has"? Ok.

**P.3, L. 5-8.It is hard to follow. Please reword or split the sentence.**

We reformulated as follow:

"Studies that reprocessed DOMINO-v2 dataset using external data usually relied either on atmospheric transport model outputs, e.g. GEOS-Chem in the Peking University OMI NO2 (POMINO) (Lin et al., 2014, 2015), or observations issued from different satellite platforms, e.g. the Cloud-Aerosol Lidar with orthogonal Polarization (CALIOP) (Castellanos et al., 2015), or even both combined together (Liu et al., 2018).".

P.3, L.7. Should be "Orthogonal" Ok.

P.4, L.21. A typo. Should be "477" Ok, done.

P.5, L.9. Remove "x" Removed.

P.6, L.4. and elsewhere, e.g. P.7, L.28; P.11, L.14; P.14, L.12. Should be subscripts. Ok.

P.10, L9. "One the main". "of" is missing. Added.

P.10, L.17. A typo in "information" Corrected

P.11, L.33. "attenuate the biases". Do you mean "reduce"? Yes. Corrected.

P.13, L.33-34. It may be hard to understand this sentence. Please clarify.

This is reformulated as follow: "The main identified reason was a reduced shielding effect applied by the effective cloud parameters resulting from a higher effective cloud pressure (cp = 350 hPa): i.e. the Lambertian reflector was defined at a lower altitude.".

P.13, L.34. "This may of course be ...". What is "this"?

We refer here to the reduction of the shielding effect mentioned in the previous sentence. We clarified accordingly.

P.16, L.29. Remove period after "scene" Removed.

P.16,L.31. Should be "potential"?

Yes, corrected.

P.17, L.4. NO2 vertical column "degradations are more important". Please reword. We rephrased by "biases are higher".

P.17, L.18. Should be 550 nm? Yes, corrected.

P.17, L.25-26. You may want to reword "the transmission of the clear fraction of the pixel through the IPA assumption". Otherwise, it is not clear.

We added the reference to the Sect. 2.2. where details were already given for a comprehensive explanation.

P.30. Fig. 4 and elsewhere in other figures. The Greek symbol "tau" in Fig. 4a is misspelled. Corrected.

P.37. Figure caption. Second (a) should be (b). Corrected.

---

## Author Response (AR2)

**Interactive discussion on AMTD-2017-286 "Minimizing aerosol effects on the OMI tropospheric NO$_2$ retrieval – An improved use of the 477 nm O$_2$-O$_2$ band and an estimation of the aerosol correction uncertainty"**

Julien Chimot *et al.*

Julien.Chimot@eumetsat.int

We would like to thank the Associate Editor for his feedbacks and suggestions. Below we address them one by one (Associate Editor comments in blue, author and co-authors in black).

Dear authors,

Thank you for submitting your revised manuscript. After reading it through together with your Response to Reviewers, I felt that you did a good job in the Response but that a couple of points were not fully accounted for in the revised text. These are things that you explain well in the Response to Reviewers but could benefit from adding an additional sentence or two in the manuscript, in case readers have similar thoughts. I therefore request the following minor additions, which I would quickly review before accepting the manuscript for publication in AMT. Specifically, these are:

1. The reviewers questioned the choice of the Henyey-Greenstein (HG) phase function. In your response you explained how this was evaluated in your previous 2016/2017 papers, which I think is reasonable, but I'd like to see this more clearly in the text. Could you add a sentence or two mentioning that HG has limitations but was evaluated for this purpose in the 2016/2017 papers somewhere probably early in section 2.3?

We added the following in Sect 2.3.:

"The HG phase function is known to have some limitations compared to more physical models. Nevertheless, it was consciously chosen in Chimot *et al.* (2017) as the main motivation has been as exploratory development of an ALH retrieval algorithm, using the OMI 477 nm O$_2$-O$_2$ absorption band, has been the aerosol correction in the visible spectral range in view
30 of tropospheric NO$_2$ retrieval. Chimot *et al.* (2016) quantitatively demonstrated that, for such a purpose, τ and ALH are the key parameters needed. Other aerosol parameters, that are more related to their optical properties, shape, and size are of a second importance. This is supported by a significant number of additional studies (Boersma *et al.*, 2004; Leitão *et al.*, 2010; Castellanos *et al.*, 2015). The main reason is that aerosol correction needs the length of the average light path in presence of scattering and absorbing particles. This is primarily driven by τ and ALH (in addition to the shape of the NO2 vertical profile), much less by the detailed properties of particles. Consequently, other details describing the shape of the scattering phase function are of second importance, even if not negligible. Moreover, areas impacted by heavy NO$_2$ are generally dominated by fine spherical particles, weakly absorbing (*e.g.* sulfate, and nitrate) or strongly absorbing (*e.g.* smoke) like in East China, South America, and Russia areas with scenes including urban, industrial, and biomass burning pollution events and for different seasons Chimot *et al.* (2017, 2018). Spheroid particles such as dust are sometimes mixed but do not overall dominate.

The HG function is known to be smooth and reproduce the Mie scattering functions reasonably well with g = 0.7 for most of aerosol types, especially for spherical particles (Dubovik *et al.*, 2002). A similar approach is considered for the operational ALH retrieval algorithms for Sentinel-4 and Sentinel-5 Precursor (Leitão *et al.*, 2010; Sanders *et al.,* 2015; Colosimo *et al.,* 2016; Nanda *et al.*, 2017), and when applying various explicit aerosol corrections in the tropospheric NO$_2$ AMF calculation 10 over urban and industrial areas dominated by anthropogenic pollution, for instance in east China (Spada *et al.*, 2006; Wagner *et al.*, 2007; Castellanos *et al.*, 2015; Vlemmix *et al.*, 2010)."

2. The reviewers also raised the issue of radiative closure resulting from MODIS (rather than OMI) as a source of AOD. In your Response you mention that you think it's important to keep these results in

the manuscript because as you note this possibility is not "naturally obvious". I agree with you about this and support keeping the analysis as-is, but again adding a sentence somewhere in Section 5.5 (*e.g.* strengthen the final paragraph of section 5.5 a little) or 6 noting this fact would be useful.

We added the following close to the end of Sect. 5.3.:

"Due to the differences in the OMI derived LER and the MODIS surface reflectance, it may be very tempted to select primarily both the OMI τ and ALH variables to avoid inconsistencies when correcting of aerosol effects. However, in this study, such a choice is not necessarily obvious for everyone as the accuracy of the ALH retrieval is strongly dependent on the requested prior τ (Chimot *et al.*, 2017). The most accurate OMI ALH retrievals were obtained with collocated MODIS τ , not with the derived OMI AOT. This would naturally suggest first that the combination of OMI ALH ? MODIS τ shall give the most accurate tropospheric $NO_2$ AMF. However, the apparent inconsistencies due to the different algorithms employed for each physical product are mostly observed in the present study through the discussion on the TOA radiance closure budget issue. They do not necessarily mean that the aerosol correction is less accurate.

The answer to such a problem is, in our opinion, not clear at this stage. But, given the fact that several studies prioritize the 5 application of multiple parameters from very diverse sources (models, ancillary instruments with different techniques, etc..) to satellite spectral measurements, we think that the issue of radiance closure budget should be kept in mind by the scientific community and further investigated in future research studies. At the end, an optimal trade off must be found between quality of Nv product and the weights given to the original satellite measurement."

The discussion about the TOA radiance closure issue is already briefly reminded in the Sect. 6 (cf. conclusion). Top keep it reasonably concise, we don't think it is very relevant to insist too much about it again.

What I am suggesting is essentially taking some of the rationale you provided in the Response to Reviewers, and inserting similar text into the manuscript itself, as it is likely that most readers of the final paper will not go back and read through the original peer review.

Please let me know if you have any questions about the above, and I look forward to seeing the next and probably final version.

Best wishes,

Andrew